# Learning Robust Generalizable Radiance Field with Visibility and Feature Augmented Point Representation

**Jiaxu Wang**[1], **Ziyi Zhang**[1], **Renjing Xu**[1†]
[1]Hong Kong University of Science and Technology (Guangzhou)
`jwang457@connect.hkust-gz.edu.cn,{ziyizhang,renjingxu}@hkust-gz.edu.cn`

## Abstract

This paper introduces a novel paradigm for the generalizable neural radiance field (NeRF). Previous generic NeRFs combine multiview stereo techniques with image-based neural rendering, yielding impressive results, while suffering from three issues. First, occlusions often result in inconsistent feature matching. Then, they deliver distortions and artifacts in geometric discontinuities and locally sharp shapes due to their individual process of sampled points and rough feature aggregation. Third, their image-based representations experience severe degradations when source views are not near enough to the target view. To address challenges, we propose the first paradigm that constructs the generalizable neural field based on point-based rather than image-based rendering, which we call the Generalizable neural Point Field (GPF). Our approach explicitly models visibilities by geometric priors and augments them with neural features. We propose a novel nonuniform log sampling strategy to improve both rendering speed and reconstruction quality. Moreover, we present a learnable kernel spatially augmented with features for feature aggregations, mitigating distortions at places with drastically varying geometries. Besides, our representation can be easily manipulated. Experiments show that our model can deliver better geometries, view consistencies, and rendering quality than all counterparts and benchmarks on three datasets in both generalization and finetuning settings, preliminarily proving the potential of the new paradigm for generalizable NeRF.

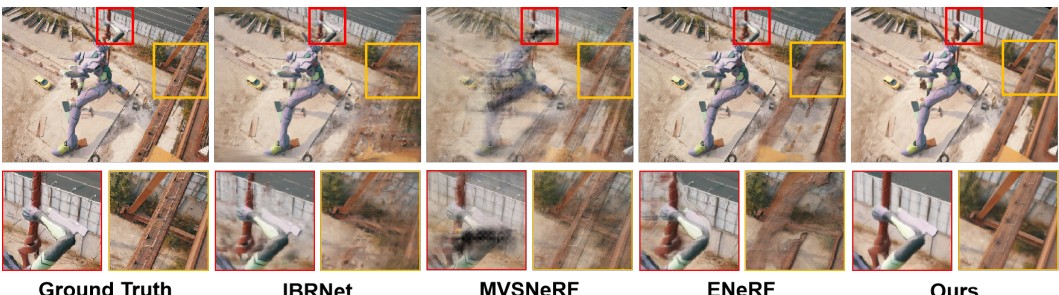

| Ground Truth | IBRNet | MVSNeRF | ENeRF | Ours |

Figure 1: Our approach produces sharper and clearer at discontinuous geometries in an unobserved scenario without per-scene training and synthesizes higher quality images than baselines.

## 1 Introduction

Novel view synthesis has emerged as an important and widely discussed topic in computer vision and graphics. In recent years, neural rendering techniques have made remarkable progress in this domain, with Neural Radiance Fields (NeRF) Mildenhall et al. (2021) being a prime instance. However, traditional NeRF-based approaches are heavily constrained by a long time for optimization and the inability to generalize to unseen scenarios.

Recent works such as Chen et al. (2021); Yu et al. (2021); Wang et al. (2021c); Lin et al. (2022); Liu et al. (2022); Johari et al. (2022) have investigated into generalizing NeRF to unobserved environments without retraining. At present, the common motivation of existing approaches is to integrate multi-view stereo (MVS) techniques into the NeRF pipeline. Their methods are called image-based neural rendering. In this paradigm, the image features extracted by networks are projected to the query 3D points which are generated by uniform sampling in volume ray marching and regressing their colors and densities. However, the paradigm has a few issues. First, geometry reasoning significantly relies on stereo matching between multi-view features. However, occlusions often destroy the feature consistencies, resulting in inaccurate geometry reconstruction. On the other hand, the sampled 3D points are often processed individually, which damages the local shape connections. In this case, distortions and artifacts commonly occur in shape-varying regions. Next, they use simple methods to aggregate features, which limits the expressive abilities of networks and causes undesirable degenerations in out-of-distribution scenes. Besides, such image-based rendering representation is not capable to interact with and edit.

To address these problems, this paper proposes a novel paradigm for generalizable NeRF based on point-based instead of image-based rendering, Generalizable neural Point Field. The proposed approach includes three main components, the visibility-oriented feature fetching, the robust log sampling strategy, and the feature-augmented learnable kernel. Furthermore, we present a three-stage finetuning scheme. Extensive experiments were conducted on the NeRF synthetic dataset Mildenhall et al. (2021), the DTU dataset Jensen et al. (2014), and the BlendedMVS dataset Yao et al. (2020). The results showcase that the proposed method outperforms all benchmarks with clearer textures, sharper shapes and edges, and better consistency between views. Additionally, we show examples of point completion and refinement results to illustrate its validity in improving reconstructed geometries. Moreover, we show the potential of the method in interactive manipulation.

The main contributions of this paper are as follows:

- We first propose a novel paradigm GPF for building generalizable NeRF based on point-based neural rendering. This novel paradigm outperforms all image-based benchmarks and yields state-of-the-art performance.
- We explicitly model the visibilities by geometric priors and augment it with neural features, which are then used to guide the feature fetching procedure to better handle occlusions.
- We propose a novel nonuniform log sampling strategy based on the point density prior, and we impose perturbations to sampling parameters for robustness, which not only improve the reconstructed geometry but also accelerate the rendering speed.
- We present the spatially feature-augmented learnable kernel as feature aggregators, which is effective for generalizability and geometry reconstruction at shape-varying areas.

## 2 RELATED WORK

**Neural Scene Representations**. Different from traditional methods that directly optimize explicit 3D geometries, such as mesh Liu et al. (2020), voxelSitzmann et al. (2019), and point cloudLiu et al. (2019), recently the use of neural networks for representing the shape and appearance of scenes Sitzmann et al. (2019); Xu et al. (2019); Tancik et al. (2020); Jiang et al. (2020) is prevailing. More recently, NeRFMildenhall et al. (2021) achieved impressive results. The following works improve NeRF in different aspects Yang et al. (2022b); Deng et al. (2022). Fridovich-Keil et al. (2022) store neural features in voxels. Xu et al. (2022) integrate point cloud into neural rendering. Yariv et al. (2021) and Wang et al. (2021b) apply NeRF to model signed distance fields. 3D Gaussian splatting Kerbl et al. (2023) achieves fast and high-quality reconstruction for per-scene settings.

**Generalizable Neural Field**. Typical NeRF requires per-scene optimization, and cannot be generalized to unseen scenes. In recent years, many methods Chen et al. (2021); Johari et al. (2022) have proposed generalizable neural fields from multi-view images. PixelNeRF Yu et al. (2021) and IBRNet Wang et al. (2021c) regard multiple source views as conditions, query features from them and perform neural interpolation. MVSNeRF Chen et al. (2021), GeoNeRF Johari et al. (2022) and ENeRF Lin et al. (2022) incorporate Multiview stereo into NeRF to generate neural cost volume to encode the scene. To avoid occlusion in stereo matching, NeuralRay Liu et al. (2022) infers occlusions in a learnable fashion. The above methods either require source views to reconstruct the neural volume before each rendering or are fully inaccessible. We propose the generalizable point-based paradigm to avoid issues that can hardly be solved by image-based methods.

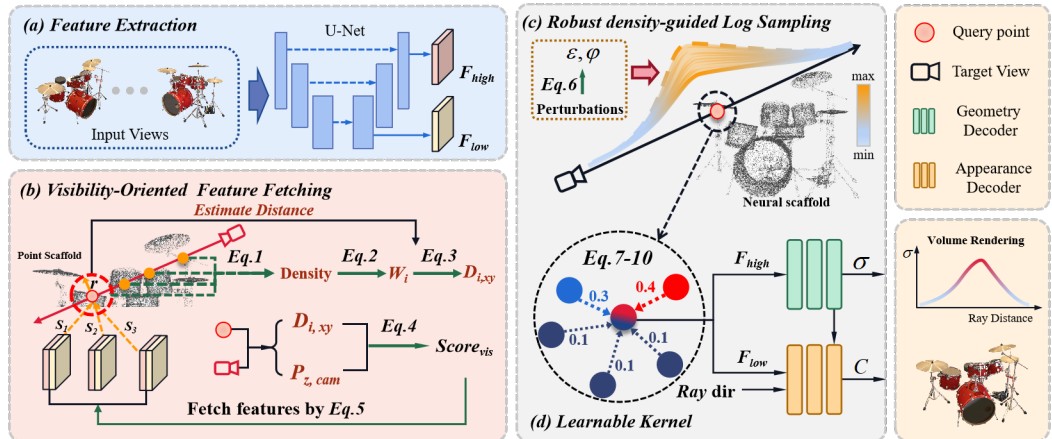

Figure 2: The Overview pipeline with our model. (a) depicts the hierarchical feature extraction. (b) is visibility-oriented feature fetching. (c) denotes the density-guided robust log sampling. (d) illustrates feature aggregation by the feature-augmented learnable kernel.

**Editing NeRF**. Most of the works (Yuan et al., 2022; Sun et al., 2022; Jiang et al., 2022) combine typical NeRF with different explicit representations. CageNeRF (Peng et al.) incorporate controllable bounding cages into NeRF. Xiang et al. (2021); Yang et al. (2022a); Kuang et al. (2022) decompose appearance editing of NeRF. Manipulation is a by-product of the GPF that can be manipulated by users in both the appearance and geometry individually.

**Point-based Rendering** Point-based rendering involves generating images from point clouds. In the past years, neural-based point renderers Rusu & Cousins (2011).Debevec et al. (1998) have made significant advancements in generating images from point clouds. Very recently, studies Dai et al. (2020); Rückert et al. (2022) have combined it with the prevailing NeRF pipeline to achieve better results, such as PointNeRF Xu et al. (2022) and Point2Pix Hu et al. (2023). Our method is partially similar to this category, while with good generalizability and performance.

## 3 METHODOLOGY

Given multi-view images, our aim is to generate representations for different source scenes without per-scene optimization. For each scene, first, we initialize the point scaffold by the MVS technique. Next, we propose to explicitly compute the physical visibilities of points to each source image. Then, we fetch image features for all points oriented by the visibility scores to mitigate occlusion effects. Up to this step, the occlusion-aware neural point representation is built from multi-view images. We access this representation by volume rendering to synthesize novel views. Furthermore, we propose a robust log sampling strategy based on the physical visibility scores, which improves not only the rendering speed but also the quality of reconstructed geometries. Last we propose to aggregate neural features for query points via a spatially learnable kernel function to obtain better generalizability. The whole paradigm is described in Fig. 2.

### 3.1 VISIBILITY-ORIENTED FEATURE FETCHING

First, we extract multi-level image features from all input source views $\{I\}_{i=1}^{n}$ via a symmetric U-Net-like Convolutional Network. As described in Fig. 2, the multi-view images are first fed to the UNet-like extractor to produce low ($F_l \in R^{H \times w \times 8}$) and high ($F_h \in R^{H/4 \times w/4 \times 32}$) scale features, respectively. Meanwhile, the PatchmatchMVS is used to initialize point cloud scaffolds.

Moreover, we fetch these features to the point cloud based on their physical visibilities. This is necessary to mitigate the adverse effect of stereo-matching inconsistency caused by occlusions. We first estimate the visible depth map for each camera viewpoint. We do not use either the depth map obtained by the MVS network or directly project points into view planes by camera parameters. Because the former contains lots of noise and the latter only produces sparse depth maps with a mass of holes. Instead, inspired by volumetric rendering, we propose a novel method to extract clean and dense depth prior from the point cloud. In detail, we emit rays from all pixels and sample 128 query points along each ray. We search neighbors in their adjacent areas with the radius $r$ and

filter out queries with no neighbors. Then we define the spatial density of each query point as the function of their neighbors, in the following:

$$d = \frac{1}{N} \sum_n exp(-\frac{1}{2}(x - x_0)^2) \tag{1}$$

where $N$ refers to the number of neighbors and $x_0$ is the coordinates of centers. The transmittance can be defined as

$$T_n = exp(\sum_{i=0}^{n} 2d_i r) \tag{2}$$

where $d_i$ is the density and r is the search radius. As to the above, the opacity weight of each query point is determined as $w_i = T_i d_i$. The depth can be estimated softly:

$$D = \frac{1}{\sum_{i=1}^{N} w_i} \sum_{i=1}^{N} w_i z_i \tag{3}$$

Next, the physical visibility score can be calculated by Eq. 4, where $P_z$ denotes the z value of the camera coordinates and $D_i$ can be considered as the visible depth estimation, especially efficient when the number of points causes bleeding problems via rasterization. Details are in the Appendix. It is natural for $P_z$ to be inherently smaller than its corresponding $D_i, xy$ by the definition of depth projection. Therefore, The visibility score is naturally constrained between 0 and 1.

$$Score = 1 - \frac{|P_z - D_{i,xy}|}{P_z} \tag{4}$$

Each point only fetches features from images with the top k visibility scores by $F_{l,n} = \sum_{i=1}^{k} w_i^l * f_{l,i}$, $F_{h,n} = \sum_{i=1}^{k} w_i^h * f_{h,i}$ and $F_{c,n} = \sum_{i=1}^{k} w_i^c * I_i$, where $w_i$s are computed over the top k visibility scores via two independent learnable modules conditioned by the physical visibilities and features, $F_l, F_h, F_c$ are low-level features, high-level features, and colors for each point.

$$F_c, F_l, F_h = \Omega(f_{l,i\in[1,k]}, f_{h,i\in[1,k]}, I_{i\in[1,k]}, v_{i\in[1,k]}) \tag{5}$$

The $\Omega$ depicts the entire feature fetching process, which is also determined by their visibility scores, moreover, we augment the process with neural features, a detailed description is in the Appendix.

## 3.2 ROBUST DENSITY-GUIDED LOG SAMPLING STRATEGY

Conventional sampling strategy either skips surfaces or requires more sampled points hence improving computational burden, which not only slows the rendering but also severely affects reconstructed geometries. Point clouds contain rich geometric priors that can be used to improve the sampling strategy. We reutilize the visible depth map introduced in Sec. 3.1 to determine the intersection points of a ray with the surface, which we call central points. In contrast to the uniform sampling used in NeRF, we sample points along a ray nonuniformly and irregularly by spreading out from the central point, following the guidance of Eq. 6.

$$\{p\}_i^{2N_k} = \left\{ P_c \pm base^{\frac{N_k}{N_k-1}*(i-1)} \right\}_i^{2N_k} \tag{6}$$

in which the total number of sampled points on a ray is $2N_k$. The $P_c$ refers to the central point, and the $base$ represents the logarithm base value that controls the sparsity of the sampling process. This equation describes that we sample points on both sides of the center in a symmetric and nonuniform way. The implication of this sampling strategy is to sample more points near the surface with rich geometric information and sample fewer points far away from it. In addition, to alleviate the influence of the error between the estimated and the real depths, small perturbations $\delta$ and $\epsilon$ are added to $base$ and $P_c$. The $P_c$ and $\delta$ in Eq. 6 are replaced by $\hat{P_c} = P_c + \delta$ and $\hat{base} = base + \epsilon$, respectively. This is also able to avoid model trapping in the local minimum. Moreover, previous generalizable NeRF methods require two identical models to implement "coarse to fine" sampling, but our model generates good sampling directly from the geometry prior and does not need the "coarse stage". Hence, we consume fewer memories and less time.

### 3.3 POINT-BASED NEURAL RENDERING WITH FEATURE-AUGMENTED LEARNABLE KERNELS

In the above two subsections, we discuss building the neural augmented point scaffold and the log sampling strategy. In this section, we introduce how to convert information from the neural scaffold to the sampled points. First of all, we search K nearest neural points for each query point within a fixed radius $r$, and filter out those without neighbors. Popular aggregation approaches in point-based rendering include inverse distance weighting, feature encoding, and predefined radial basis functions. They suffer from distortions and blurs when generalizing to varying geometries. Because geometries and scene scales drastically vary across different scenes. Thus bridging neural scaffolds and the continuous field via simple predefined functions is challenging. Therefore, we propose a feature-augmented learnable kernel to aggregate features. For the nearest K points $\left\{p_i^1...p_i^K\right\}$ of the query $p_i$, we first compute their spatial information encoding including the distances, positions under camera coordinate systems, and relative coordinates by Eq. 7, detailed in Appendix.

$$h_{s,i}^k = MLP(cat(p_i^k, p_i^k - p_i, ||p_i^k - p_i||_1)) \tag{7}$$

Then we input the local spatial features and their neural features ($F_*$ in Eq. 5, $*$ contains $l, h, c$) to the feature-augmented learnable kernel, which is illustrated by Eq. 8 to Eq. 10.

$$\hat{w}_i^k = MLP(h_{s,i}^k), \tag{8}$$

$$v_i^k, H_i^k = Sigmoid(h_v), ReLU(h_H) = MLP(cat(h_{s,i}^k, F_*)), \tag{9}$$

$$F = \sum_{k=1}^{K} softmax(v_i^k * \hat{w}_i^k) H_i^k, \tag{10}$$

where $cat$ is the concatenation operation, the spatial features are first initialized as weights $\hat{w}_i^k$. Meanwhile, they are then concatenated with the neural features to produce another feature vector $H_i^k$ and a tunning coefficient. The tuning coefficients are multiplied with the $\hat{w}_i^k$ to adjust their ratio, and then $H_i^k$ is weighted summation to obtain the final feature vector for each query point. The results are interpreted as $c$ or $\sigma$ by decoders. Besides, if the final target is the color, $F_*$ denotes the concatenation of $F_l$ and $F_c$ in Eq. 5. In contrast, if the target is $\sigma$, it should be $F_h$. With this feature-augmented learnable kernel, our network expresses local features better and performs well at geometric discontinuities, such as the edge of an object.

Finally, conventional volume rendering ($c = \sum_K T_j(1 - exp(-\sigma_j\delta_j))c_j$) is applied to obtain color at pixels, where $T_j = exp(-\sum_{t=1}^{j-1}\sigma_j\delta_j)$. The only loss we used to train our model is the MSE loss between predictions and groundtruth, which is depicted as $L_c = \frac{1}{|R|}\sum_{r \in R} ||\hat{c}_r - c_{gt}(r)||$.

### 3.4 HIERARCHICAL FINETUNING

We present a hierarchical finetuning paradigm. Different from previous image-based generic NeRF approaches, we remove the image feature extractor before fintuning, and only maintain features attached to the point scaffold, because our method can render views independently of source images.

The proposed hierarchical finetuning paradigm includes three stages: 1. feature finetuning 2. point growing and pruning 3. point refinement. The first stage optimizes neural features stored in the point scaffold and the weights of the learnable kernels. The initialized point clouds sometimes are imperfect and contain artifacts and pinholes. Inspired by PointNeRF, we introduce point completion and pruning techniques in the second finetuning stage. For point completion, we grow new points at places with higher opacity ($\alpha = 1 - exp(-\sigma_j\delta_j)$). If the opacity of a candidate point is larger than a preset threshold $\alpha > T_{opacity}$ and the distance to its nearest neural points is not smaller than another threshold $d_{min} \geq T_{dist}$, a new point with averaged features over its neighbors is added.

Our point pruning is different from PointNeRF because our representation does not rely on the per-point probabilistic confidence. Instead, we follow the feature-augmented kernel to recompute the opacity at the positions of neural points and delete them when they have smaller opacity.

The third stage is called point refinement. Point positions might be suboptimal in their vicinities. In this stage, we freeze features and all networks and iteratively adjust the coordinates of the point scaffold, following Eq. 11. In detail, a trainable offset $\Delta p_i$ is assigned to each point, and $v$ refers to the camera viewpoint. The offset is regularized with its L2 loss to ensure that they would not move

| Training Setting | Methods | NeRF Synthetic | | | DTU | | | BlendedMVS | | |
|---|---|---|---|---|---|---|---|---|---|---|
| | | PSNR↑ | SSIM↑ | LPIPS↓ | PSNR↑ | SSIM↑ | LPIPS↓ | PSNR↑ | SSIM↑ | LPIPS↓ |
| Generalization | IBRNet | 26.91 | 0.925 | 0.113 | 25.17 | 0.902 | 0.181 | 22.01 | 0.813 | 0.171 |
| | MVSNeRF | 23.65 | 0.827 | 0.181 | 23.50 | 0.818 | 0.314 | 20.27 | 0.795 | 0.290 |
| | ENeRF | 26.69 | 0.947 | 0.097 | 25.77 | 0.913 | 0.168 | 21.88 | 0.758 | 0.176 |
| | Neuray | 27.07 | 0.935 | 0.085 | 25.94 | 0.925 | 0.122 | 23.24 | **0.878** | 0.106 |
| | **Ours** | **29.31** | **0.960** | **0.081** | **27.67** | **0.945** | **0.118** | **26.65** | **0.878** | **0.098** |
| Finetuning | IBRNet | 29.95 | 0.934 | 0.079 | 28.91 | 0.908 | 0.130 | 25.01 | 0.721 | 0.277 |
| | MVSNeRF | 28.69 | 0.905 | 0.160 | 26.41 | 0.881 | 0.274 | 21.11 | 0.796 | 0.279 |
| | ENeRF | 29.21 | 0.931 | 0.077 | 28.13 | 0.931 | 0.101 | 24.81 | 0.863 | 0.201 |
| | Neuray | 30.26 | 0.969 | 0.055 | 29.15 | 0.935 | 0.104 | 26.71 | 0.885 | 0.083 |
| | **Ours** | **33.28** | **0.983** | **0.037** | **31.65** | **0.969** | **0.081** | **28.78** | **0.944** | **0.073** |

Table 1: Quatatitive comparisons on the three datasets under generalization and finetuning settings. The PSNR, SSIM, and LPIPS are computed.

too far from their original positions. The three stages experience iterative optimizations one after another until the loss no longer decreases.

$$\Delta p_i = argmin(\sum_v ||I_\theta(v|p_i + \Delta p_i) - I_{gt}||_2^2 + \sum_{i=1}^{N} ||\Delta p_i||^2) \qquad (11)$$

## 4 EXPERIMENT

**Datasets**. We pretrain our model on the train set of DTU Yao et al. (2020), in which we follow the train-test split setting introduced in MVSNeRF. To evaluate the generalization ability of our model, we test the pretrained model on NeRF Synthetic Dataset Mildenhall et al. (2021), the test set in DTU Dataset and large-scale scenes in BlendedMVS Yao et al. (2020).

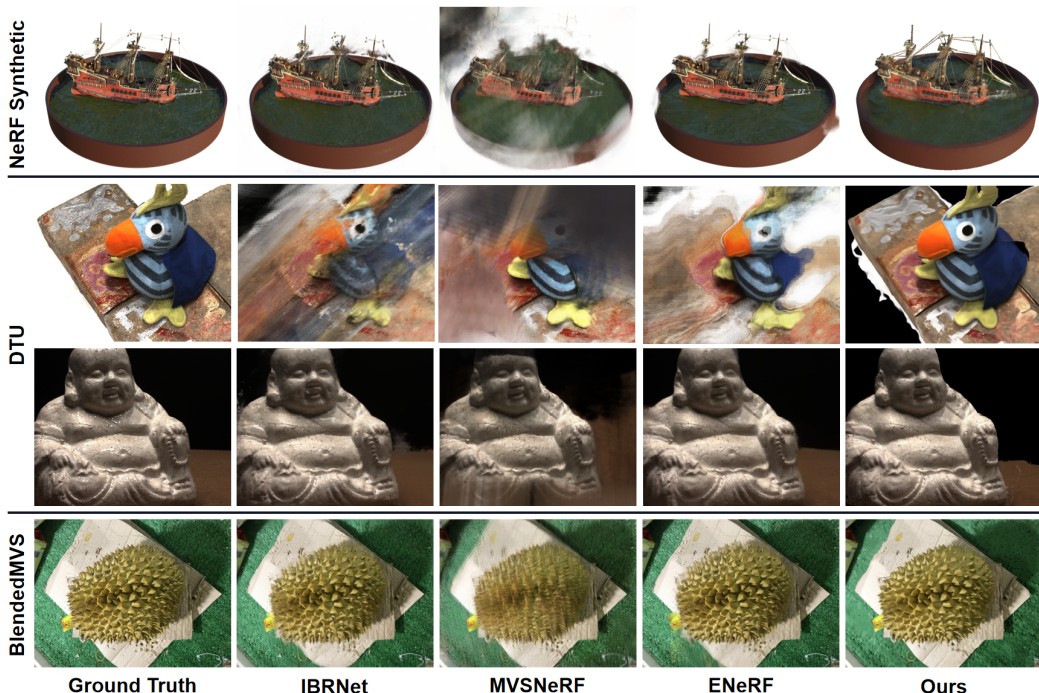

Figure 3: Qualitative comparisons of novel view synthesis under generalization setting.

### 4.1 COMPARISON WITH BASELINES

We compare the proposed method with IBRNet, MVSNeRF, and ENeRF, all of which are recent state-of-the-art open-source generic radiance field methods, in both the generalization setting and finetuning setting. All generalization methods are pretrained on the same training scene. The qualitative and quantitative results are reported in Table 1 and Fig. 3 and 4 respectively. In Table 1, we list the metrics of PSNR, SSIM, and LPIPS on the three datasets. Our method outperforms all other methods in both generalization and finetuning settings by a considerable margin. Fig. 3 vividly

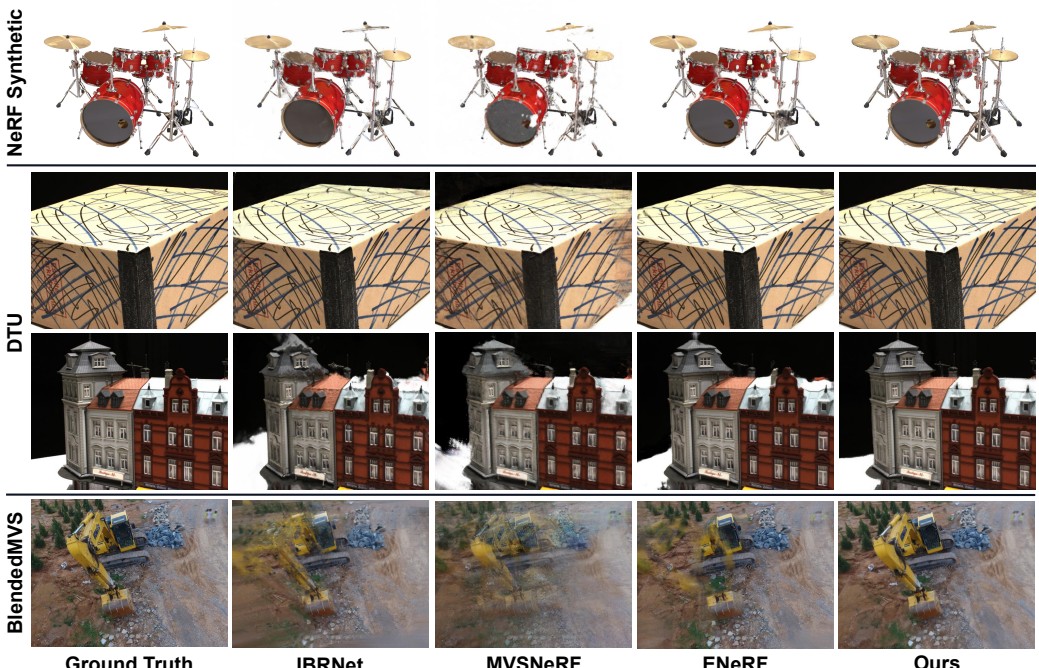

Figure 4: Qualitative comparisons of novel view synthesis under finetuning setting.

| Training Setting | Methods | NeRF Synthetic | | | DTU | | |
|---|---|---|---|---|---|---|---|
| | | PSNR↑ | SSIM↑ | LPIPS↓ | PSNR↑ | SSIM↑ | LPIPS↓ |
| Finetuning | PointNeRF | 30.71 | 0.961 | 0.081 | 28.43 | 0.929 | 0.183 |
| | Point2Pix | 25.62 | 0.915 | 0.133 | 24.81 | 0.894 | 0.209 |
| | **Ours** | **33.28** | **0.983** | **0.037** | **31.65** | **0.970** | **0.081** |

Table 2: Quantitative comparisons with point-based methods on NeRF Synthetic and DTU datasets.

depicts that other models produce severe artifacts at places with local shape variations, especially edges of the object. Existing methods cannot handle extremely varying local shapes because they only consider independent sampled points. Instead, our model fully utilizes the geometric prior and presents the learnable kernels to better abstract local features.

In the finetuning setting (Fig. 4), it is observed from DTU results that the other three models incorrectly infer the geometry, caused by occlusions in source views. Furthermore, in the Blended-MVS example, the other three models completely collapsed. This is because the source images are distributed in a scattered manner, and occlusions exist between adjacent views, which damage the image-based methods. By contrast, our model leverages visibility-based feature fetching to effectively address the issue. Even if the novel view lacks enough nearest source views, our model can generate plausible high-quality images. We also show the quantitative comparison in Table 4.1 between our approach with other point-based rendering methods, PointNeRF Xu et al. (2022) and Point2Pix Hu et al. (2023) (Point2pix is our unofficial implementation). The results show our method still has significant advantages. More detailed comparisons are depicted in the Appendix.

## 4.2 COMPARISON WITH NEURAY

We individually report the comparisons with Neuray in this subsection because it implicitly models the occlusion by a trainable visibility module. However, it performs poorly in out-of-distribution scenarios. In contrast, we compute the visibility in an explicit way and slightly adjust it based on neural features. Our method yields better geometries and consistencies between views, which can be seen in Fig. 5. It is observed that NeuRay generates blur and unsharp geometries, resulting in incorrect reconstructions in local areas with drastically varying shapes. Nevertheless, our method produces clean and sharp geometrics even though at places with geometric discontinuities. Furthermore, we also compute the metrics of NeuRay on the three datasets, which we report in Table 1 as

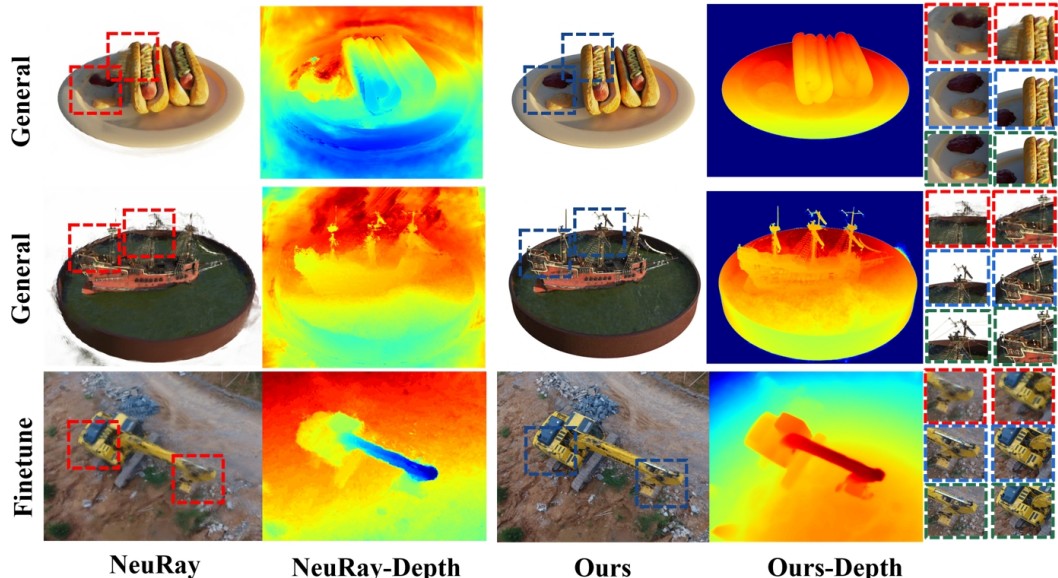

Figure 5: Qualitative Comparisons between ours and the implicit occlusion-aware image-based rendering method: NeuRay, including the rendering views and the reconstructed geometries. The green box is the groundtruth at the right side of the figure.

| Methods | Training Time ↓ | Rendering Time ↓ | PSNR ↑ | SSIM ↑ | LPIPS ↓ |
|---|---|---|---|---|---|
| Uni 64 | 22h | 86.6s | 21.42 | 0.776 | 0.312 |
| Uni 128 | 26h | 103.2s | 25.08 | 0.878 | 0.382 |
| Uni 64+128 | 29h | 124s | 26.56 | 0.903 | 0.179 |
| Surf 2 | 11.5h | **0.62s** | 24.10 | 0.891 | 0.226 |
| log16-no$\epsilon$ | 8.5h | 1.04s | 27.13 | 0.927 | 0.136 |
| **log16(ours)** | **8.5h** | 1.04s | **27.67** | **0.945** | **0.118** |

Table 3: Ablations about different sampling strategies. Uni+number refers to uniform sample "number" points. Uni 64+128 denotes the "coarse to fine" strategy used in conventional NeRF. Surf2 follows the method in ENeRF, only sampling 2 points around the surface. log16-no$\epsilon$ is the log sampling without perturbation. log16(ours) refers to the full setting of log sampling.

well. Obviously, NeuRay delivers more remarkable results than the other three benchmarks thanks to its implicit occlusion modeling. However, it suffers from degeneration when generalized to out-of-distribution scenes. Besides, compared with our explicit visibility modeling, implicit modeling causes severe damage to view consistencies, which are shown in our supplementary video. By contrast, our model gives preferable reconstruction qualities on both geometries and novel views under both generalization and finetuning settings.

### 4.3 ABLATION STUDIES AND ANALYSIS

**Main components**. This subsection presents the effectiveness of the other main components of our method, including the robust log sampling and the feature-augmented learnable kernel. We experimentally prove the necessity of the irregular log sampling strategy in terms of performance and time by comparing it with uniform and surface sampling methods. Table 3 illustrates that the novel sampling strategy delivers better performance while consuming less time. A visual aid is provided in Sec. E of Appendix. Even though the rendering speed of Surf2 used in ENeRF is slightly less than ours, its performance is largely degraded because the estimated surface is not accurate thus sampling only 2 points around it leads to dramatic errors. The metrics are evaluated on the DTU test set under generalization settings, we also present visual aids to facilitate understanding in Appendix Sec. E.

Second, we compare our feature-augmented spatial learnable kernel with inverse distance weighting Xu et al. (2022), Gaussian radial basis kernel Abou-Chakra et al. (2022), and trainable feature

| Methods | Inverse distance weights | Radial basis kernel | Trainable encoding | Ours |
|---|---|---|---|---|
| $PSNR_g\uparrow$ | 23.61 | 24.75 | 25.20 | 27.67 |
| $PSNR_{ft}\uparrow$ | 28.18 | 25.88 | 28.92 | 31.65 |

Table 4: Quantitative Comparisons with different feature aggregators. "g" refers to generalizable setting and "ft" denotes finetuning.

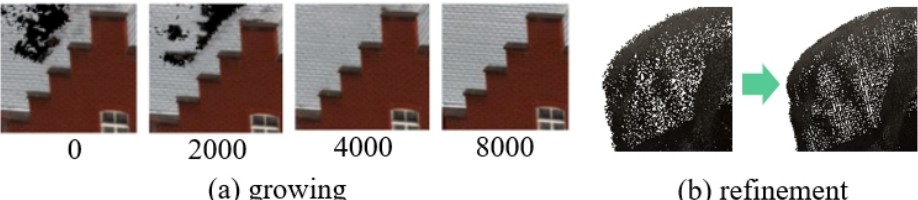

(a) growing          (b) refinement

Figure 6: Instances of point completion and refinement in finetuning.

encoding Wang et al. (2021c) for feature aggregation. Table 4 quantitatively gives the comparison results. It is noted that the inverse distance weights yield the worst quality in generalization, whereas significantly improved after finetning. The predefined kernel function produces good results in generalization but does not benefit so much from finetuning. However, our method produces the best results in both generic and finetuning.

**Finetune**. In addition, we evaluate our hierarchical finetuning paradigm, especially in the second and third stages. It can be observed from Fig. 6 (a). that the holes caused by imperfect initial point cloud have gradually filled up with the increasing training steps. It is clear from Fig. 6 (b) that the point refinement module enables points closer to the real object surface, which effectively benefits both the density-guided log sampling procedure and the reconstructed geometry.

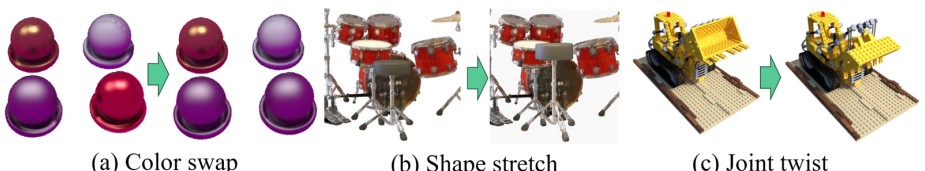

(a) Color swap          (b) Shape stretch          (c) Joint twist

Figure 7: Interactive manipulation on appearance and geometry

**Interactive and Editable**. Our approach also comes with a good by-product compared with currently popular image-based generic NeRF models. Our representation is easy to interactively manipulate. Here we simply conduct some preliminary attempts to show its potential for manipulation. For a well-deployed neural point scaffold, we use open-source 3D software, such as Blender, to manipulate the positions of points and maintain the features stored in them. The results are shown in Fig. 7. The object in scenes can either move to new places (a) or slightly change its shape (b). Besides, it can also be handled under physical constraints (c).

## 5  DISCUSSION

**Conclusion**. This paper proposed a novel point-based paradigm for building generalizable NeRF. This model explicitly models physical visibility augmented with features to guide feature fetching, which better solves occlusions. Besides, a robust log sampling strategy is proposed for improving both reconstruction quality and rendering speed. Moreover, we present the spatially feature-augmented learnable kernel to replace conventional feature aggregators. Thanks to this, the performance at geometric discontinuities is improved by a large margin. Experiments on the DTU, NeRF, and BlendedMVS datasets demonstrate that our method can render high-quality plausible images and recover correct geometries for both generic and finetuning settings.

**Limitation**. Our GPF requires the PatchmatchMVS to initialize the point scaffold in the generalization stage. In the future, we plan to propose a neural-based initialization module that can be jointly trained with our modules simultaneously.

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

# Appendix

## A    MODEL ARCHITECTURES

In this section, we complementally introduce the architectures of the visibility-oriented feature fetching module and our low and high-level feature aggregators.

### A.1    DETAILED DESCRIPTIONS OF THE VISIBILITY SCORES

As we explained in the main body of this paper, we first estimate the depth maps by the proposed point density-based method. Notably, z-buffer point rasterization or splatting methods can also be used as long as the number of points is sufficient to avoid the "bleeding" problem because the geometry prior is just estimated and will be compensated by the following neural augmented module.

Then we precompute physical visibility scores of each point to all source images, as Fig. 8 states. We use Fig. 9 to help us clearly depict the physical meaning behind the design of the visibility scores. In this figure, in the camera coordinate system, $P_z$ represents the z-value of a specific point. It indicates the distance between the projection of the line connecting the point and the camera center to the camera's optical axis. The value $D_{i,xy}$ is obtained by interpolating the depth map using Eq. 1 to 3. If a point, such as point A in the figure, is visible from the viewpoint, it implies that it lies on the object's surface. In this case, the z-value of the point in the camera coordinate system should roughly match the interpolated depth value, $D_{i,xy}$. On the contrary, if a point (e.g., point B) is not visible from the viewpoint, its z-coordinate can only be greater than $D_{i,xy}$. There should not be any other points between $D_{i,xy}$ and the camera's center, as it would cause a change in the value of $D_{i,xy}$ accordingly. Consider another case where a point lies outside the viewing angle's frustum, and its z-value is smaller than the minimum depth on the depth map, as illustrated by point C in the figure. In this case, the point would be projected outside the image plane. However, during interpolation on the plane, we employ zero padding. Therefore, the value of $D_{i,xy}$ for this point would be zero, which is smaller than $P_z$ as well.

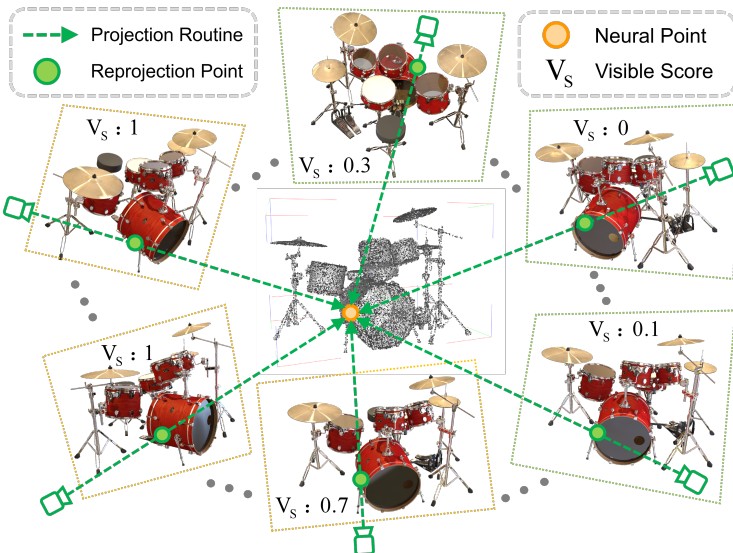

Figure 8: Schematic diagram of physical visibility score computing. We precompute visibility scores for each independent point to all source images. $V_s$ refers to the score which ranges from 0 to 1, larger score represents more possibility of this point to be viewed from the associated view.

### A.2    DETAILED IMPLEMENTATION OF FEATURE AGGREGATORS

When the visibility scores are obtained, we select three source views with the first three largest visibilities for each point. Next, the features on the three selected views will be aggregated by

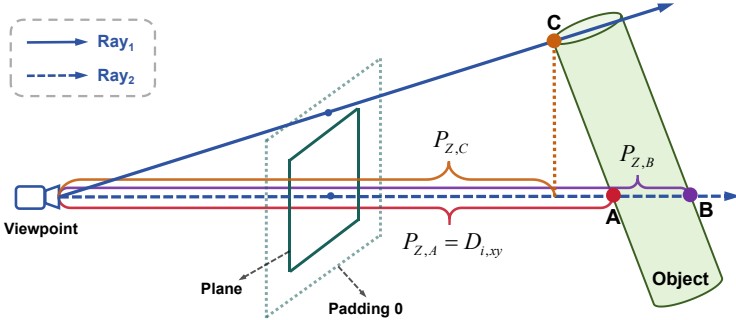

Figure 9: Visualization of the physical meaning of the visibility score.

our proposed attentive visibility module, instead of simply averaging over them, in a more refined manner. First, the fetched low- and high-dimensional features are projected to 32 dimensional space by MLPs. Here we design two aggregators, one for low-level features and colors, the other for high-level features. Regarding the low-level feature aggregators, a multi-layer perceptron (MLP) consisting of two hidden layers is employed. This MLP takes in the concatenation of low-level features, colors, and visibilities as input and generates weights for features and colors as output (refer to Eq. 1). In the equation, the variable $v$ represents the physical visibility scores. The feature tuning weights, denoted as $w_i^l$, are utilized to make slight adjustments to the original visibility scores. The visibility scores are multiplied by the feature tuning weights and then normalized to ensure that their summation equals 1, then the weighting sum of $h_l$ is performed to obtain the final features with the consideration of visibilities, as stated in Eq. 13.

$$w_l^{i=1,\dots,k}, h_l^{i=1,\dots,k} = f(F_l^i, c_i, v^i)^{i=1,\dots,k} \tag{12}$$

$$F_l = \sum_i^k Norm(w_l^{i\in k} * v^{i=1,\dots,k})h_l^i \tag{13}$$

When it comes to aggregating the high-level features, there exists minimal disparity. The aggregator employed for high-level features also utilizes a multi-layer perceptron (MLP), albeit with distinct inputs. Initially, features are concatenated with colors akin to the low-level aggregator. Subsequently, the mean and variance are calculated across the k vectors, and the resultant values are concatenated as an additional input vector. This approach is adopted due to the inclusion of more comprehensive information pertaining to the scene geometries within the high-level feature consistency.

We here formulate a vivid diagram to depict the process of the feature-augmented learnable kernel (Eq. 8 to Eq. 10 in the main paper) in Fig. 10. Clearly, the weights to sum neural features are initially produced by spatial information in the above branch, but experience tuning with the feature augmentations of the below branch. Besides, the cross-scene decoders are two simple MLPs with

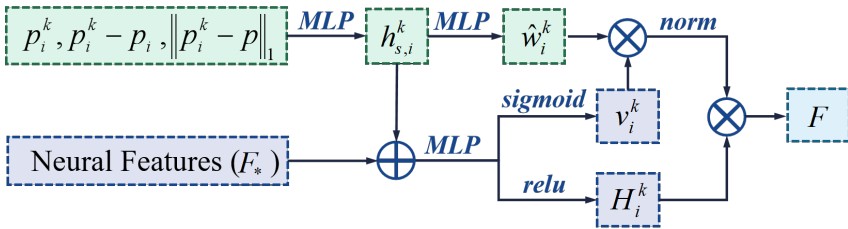

Figure 10: Schematic diagram of the feature-augmented learnable kernel.

two layers to translate neural features aggregated from their neighboring points to the density $\sigma$ and the color weights. Simple decoder architectures can enhance the representation ability of the neural features.

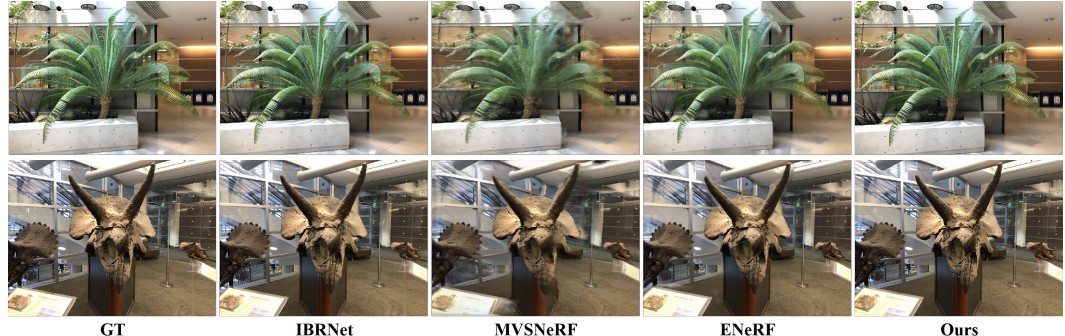

GT                    IBRNet                    MVSNeRF                    ENeRF                    Ours

Figure 11: Qualitative comparisons on LLFF dataset.

| Training Setting | Generalization | | | Finetuning | | |
|---|---|---|---|---|---|---|
| Methods | PSNR↑ | SSIM↑ | LPIPS↓ | PSNR↑ | SSIM↑ | LPIPS↓ |
| IBRNet | 25.13 | 0.817 | 0.205 | 26.73 | 0.851 | 0.175 |
| MVSNeRF | 21.93 | 0.795 | 0.252 | 25.45 | 0.877 | 0.192 |
| ENeRF | 22.78 | 0.808 | 0.209 | 24.89 | 0.865 | 0.199 |
| Neuray | 25.35 | 0.818 | 0.198 | 27.06 | 0.850 | 0.172 |
| **Ours** | **26.01** | **0.829** | **0.184** | **27.79** | **0.872** | **0.171** |

Table 5: Quantitative comparisons on LLFF dataset.

## B IMPLEMENTATION DETAILS

We train our generalizable model on a single RTX3090 GPU for 100k iterations using Adam optimizer with an initial learning rate of 5e-4 and a cosine annealing schedule with annealing $\alpha$ of 0.1. In our experiments, in the training stage, we selected 10 input views to compute the visibility and set the top-k as top-3 to perform visibility-based feature fetching. In contrast, in the test stage, we compute visibility and fetch features across all source images. In our log sampling strategy, the two perturbations are sampled from the two uniform $\epsilon \sim U(-10, 10)$ and normal $\delta \sim N(0, 0.01)$ distributions respectively. The parameters of the two distributions are determined by the scale of the scenes. The base is set to 1.35 and we sample 16 points for each ray. For each iteration, the training batch is 512. In the feature-augmented learnable kernel, the k nearest neighbors are selected as 8. Besides, we enlarged the NeRF Synthetic scene 100 times to make it in accordance with the scale of DTU training set. In finetuning, we first extract and aggregate all image features to the point scaffold. Then, source images are not required in the finetuning setting. Moreover, the finetuning is effectively fast and only consumes 50k iterations.

In the first stage of finetuning setting, the features and parameters of networks have the initial learning rate of 1e-5 but the color of each point is trained with 1e-7. In the point completion stage, notably, only if the radius of the hole is smaller than the searching radius, the point cloud can be recovered. Therefore, we slightly enlarge the search areas 1.5 to 2 times than that in the generalization stage. To extend searching areas, we reutilize uniform sampling. In point pruning, it is impossible to check all the neural points at each iteration step, thus we randomly select 3192 neural points at each pruning step. After the refinement of point positions, the precomputed visible depth map should be regenerated.

## C ADDITIONAL RESULTS AND ANALYSIS

In this section, we provide additional qualitative results under generalization and finetuning settings with enlarged details.

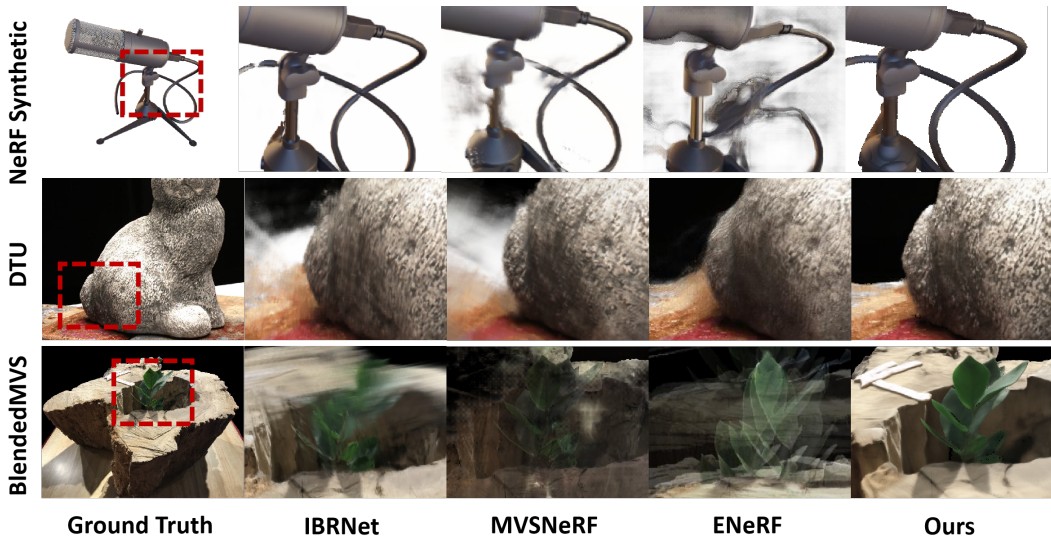

Figure 12: Additional qualitative results on the three datasets with enlarged details under generalization setting.

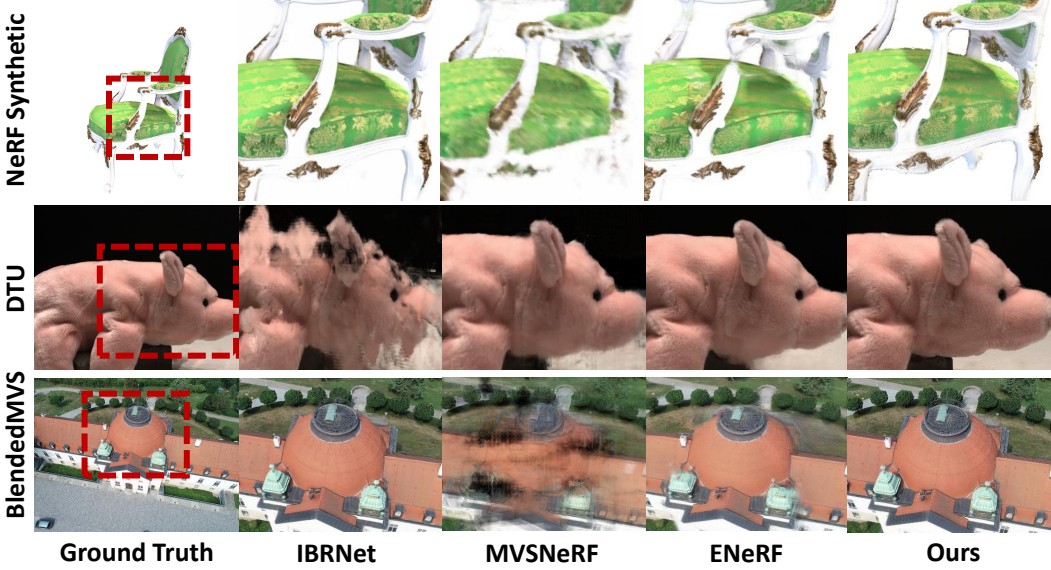

Figure 13: Additional qualitative results on the three datasets with enlarged details under finetuning setting.

## C.1 QUALITATIVE RESULTS ON THE LLFF DATASET

To provide further comparisons and evaluations, we conducted additional experiments on the LLFF dataset for all baselines and our proposed method. This section includes the qualitative results, depicted in Fig. 11, as well as the quantitative results presented in Table 5. These findings offer insights into the performance and effectiveness of our method in comparison to the baselines.

## C.2 QUALITATIVE RESULTS WITH ENLARGED DETAILS

In Fig. 12 and 13, additional results under generalization and finetuning settings with their large detail boxes are given. Our representation performs stably under all three datasets, even if the geometry is complex or the source images are not close to the target view. However, in BlendedMVS

Dataset, the camera views often distribute sparsely, thus image-based rendering cannot acquire sufficient information from a few surrounding source images. Utilizing all source images would cause computation redundancy. Therefore, the counterparts cannot deliver satisfactory results on the three Datasets. Furthermore, even if finetuning can improve the quality of rendering images of the counterparts, it cannot deal with the in-realism caused by occlusion. The chair scene in NeRF Dataset is an example of that. The back leg of the chair disappears. Moreover, the mic scene in NeRF dataset contains many geometry inconsistencies, which lead to blur and artifact of reconstruction images, our model performs better in such scenes as well.

## D  COMAPRISONS OF THE RECONSTRUCTED GEOMETRY WITH BASELINES

Fig. 5 indicates the performance of our model on the reconstruction of geometries. The predicted depth map is clear and contrasting. In this section, we report the quantitative results to further describe our benefits in Table 6.

In this table, The RMSE refers to the root mean square error computed over the normalized depth groundtruth, Acc. T denotes the accuracy of the threshold, we set the threshold as $195\%$ for all experiments. The $\text{Ours}_{wo/vis}$ refers to the variant of our model by removing the visibility scores. In addition, all tests are under generalization settings because finetuning cannot faithfully reflect the real understanding of models to the realistic geometries. The performance would see a decrease if we remove the explicit occlusion modeling, which proves the validity of the visibility-oriented feature fetching module.

| Method | | IBRNet | ENeRF | Neuray | $\text{Ours}_{wo/vis}$ | Ours |
|---|---|---|---|---|---|---|
| NeRF Synthetic | RMSE↓ | 0.677 | 0.527 | 0.547 | 0.294 | **0.161** |
| | Acc.T↑ | 0.380 | 0.159 | 0.119 | 0.596 | **0.787** |
| DTU | RMSE↓ | 0.321 | 0.435 | 0.162 | 0.189 | **0.122** |
| | Acc.T↑ | 0.896 | 0.741 | 0.911 | 0.905 | **0.936** |

Table 6: Quantitative comparisons of depth maps on NeRF Synthetic and DTU datasets.

## E  ADDITIONAL ABLATION STUDIES

Additional visualizations about ablation studies for the main components are provided in this section to further interpret Sec. 4.3 in the main paper. Besides, we report the ablations of the depth map generated by our model with different configurations. Moreover, the effect of point numbers is compared and reported. Here we first present Fig. 14 to visually help understand the differences between various sampling strategies, i.e. Table 3 in the main paper. In this figure, the x-axis corresponds to training time, the y-axis represents the PSNR, and the size of the circles indicates the rendering time for an 800*800 image. By examining the figure, we can readily observe that the log sampling method consumes the least amount of training time while achieving the highest PSNR. Additionally, it demonstrates the second-fastest rendering speed, slightly slower than the surface sampling technique.

### E.1  VISUALIZATIONS OF ABLATION STUDIES

Fig. 15 shows the detailed structure at places with dramatically varying geometries under generalization settings. Our learnable kernel reconstructs the sharpest edges with fewer blurs and fogs, whereas other methods suffer from confused appearances and disconnected shapes.

Furthermore, Fig. 16 gives visual examples to compare different sampling strategies. We can see that uniform sampling fails to maintain detailed textures in renderings, even though the number of sampled points increases to 192. However, our density-based log sampling can recover distinct appearances and unambiguous geometry.

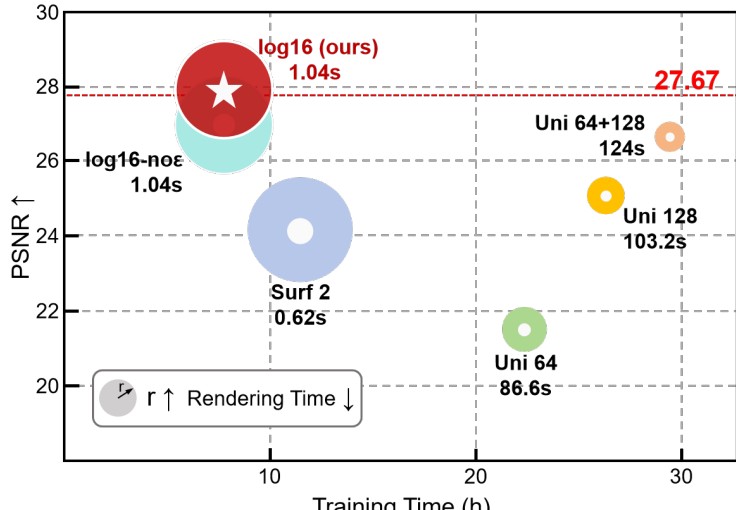

Figure 14: Ablation studies of different sampling strategies.

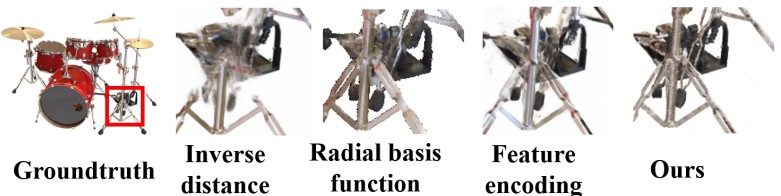

| Groundtruth | Inverse distance | Radial basis function | Feature encoding | Ours |

Figure 15: Ablation study of various feature aggregators.

## E.2 VALIDATION OF THE LOW- AND HIGH-LEVEL DECOMPOSITION

We conducted experiments on the DTU dataset to analyze the impact of different combinations of features on the rendering qualities, including only low-level features, only high-level features, and the combinations of low- and high-level features. The ablation results are listed in Table 7. Here we evaluate the rendering quality in both generalization and finetuning settings and the reconstructed depth map in the generalization setting. The results show that separating appearance and shape into individual code is indeed meaningful.

| Training Setting | Generalization | | | Finetuning | | | | |
|---|---|---|---|---|---|---|---|---|
| Methods | PSNR↑ | SSIM↑ | LPIPS↓ | PSNR↑ | SSIM↑ | LPIPS↓ | RMSE↓ | Acc.T↑ |
| Only-low | 24.15 | 0.896 | 0.178 | 30.16 | 0.921 | 0.133 | 0.383 | 0.865 |
| Only-high | 22.94 | 0.852 | 0.231 | 30.47 | 0.928 | 0.132 | 0.155 | 0.917 |
| **low-high** | **27.67** | **0.945** | **0.118** | **31.65** | **0.970** | **0.081** | **0.122** | **0.936** |

Table 7: Quantitative comparisons between the compositions of different levels of features.

## E.3 RECONSTRUCTION OF GEOMETRIES

Here we qualitatively report the depth visualization under different model settings. Fig. 17 (b) refers to the groundtruth depth, and (c) is the depth predicted by PatchmatchMVS Net Wang et al. (2021a). (d) denotes our full model prediction. (e) is the prediction of our model except for using the feature-augmented learnable kernel introduced in the main paper. (f) is also the prediction of our model but using "coarse to fine" uniform sampling rather than the proposed log sampling. From these figures, we observe that the full model provides the most reasonable prediction. Besides, the prediction of

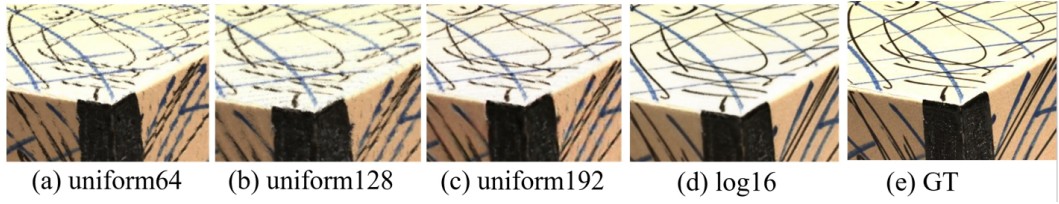

(a) uniform64    (b) uniform128    (c) uniform192    (d) log16    (e) GT

Figure 16: Ablation study of different sampling strategies.

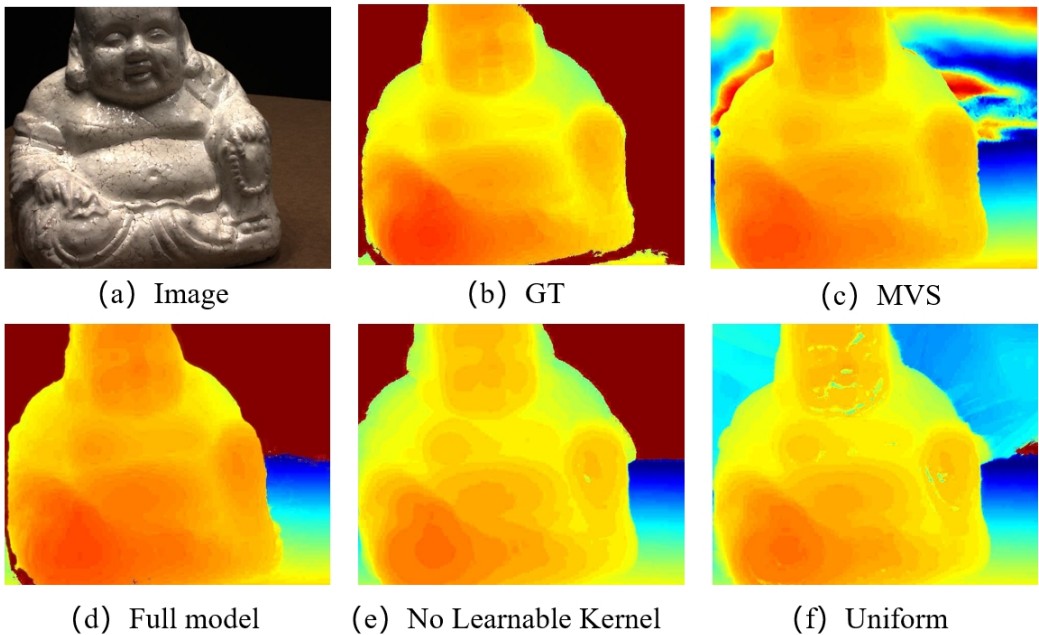

(a) Image    (b) GT    (c) MVS

(d) Full model    (e) No Learnable Kernel    (f) Uniform

Figure 17: Ablation study of depth predictions.

MVSNet is unable to handle regions without depth, which also occurs in the prediction of other generic NeRF models.

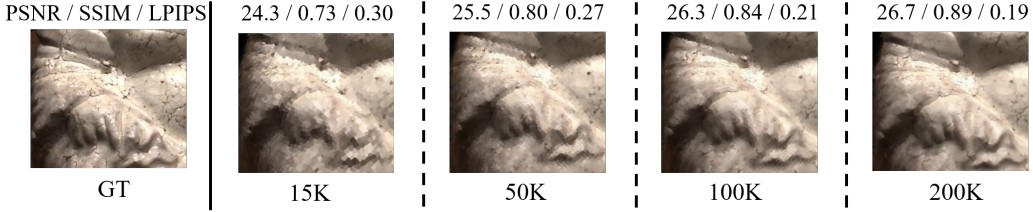

Figure 18: Ablation studies of the influence of the number of points in the point scaffold. The figure reports the rendering quality when the number of points is about 15K, 50K, 100K, and 200K respectively. The quantitative metrics including PSNR, SSIM, and LPIPS are calculated.

### E.4    INFLUENCE OF THE POINT SCAFFOLD DENSITY

We additionally analyze the effect of the different number of points in the scaffold. In detail, we test our trained model on the DTU Scan114 scene with different levels of point downsampling under the generalization setting. It is shown in Fig. 18 that the quality of rendering is slightly impacted by the density of the point scaffold. However, fewer number points lead to faster rendering speed.

Thus, this is a trade-off between rendering quality and speed. We recommend that the number of points 50K 100K is enough to produce plausible novel images with satisfactory speed for the DTU dataset. This experiment demonstrates the robustness of our proposed representation, which cannot be drastically affected by the point density.

In addition, we found that for the large scene, too fewer number of initial points would lead to smoother images due to the lack of representation of high-frequency information. Here we give an example to illustrate it in Fig. 19. Obviously, the initial number of points is insufficient for the large scene at (b) in the figure, leading to the smoothness of the image. That is intuitive since larger scenes typically require more points for initialization.

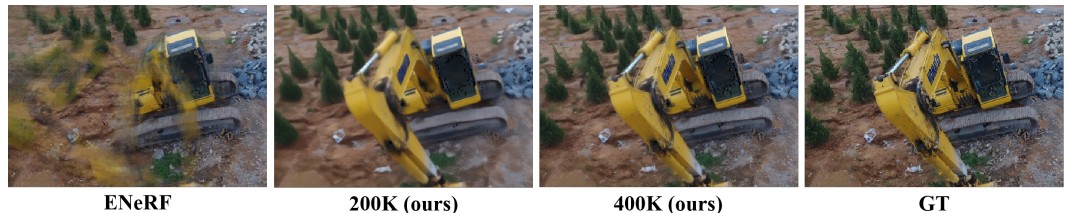

| ENeRF | 200K (ours) | 400K (ours) | GT |

Figure 19: Examples to indicate fewer points result in smoother images.

## F  MANIPULATION OF THE REPRESENTATIONS

In this paper, we also prove that the proposed representation has the potential to interact with users. In this section, we in detail introduce how to edit the generalizable point field, including moving, recoloring, and stretching, which corresponds to Fig. 7 in the main paper.

**Object movement**. If one wants to move an object in the scene, he can directly move the corresponding neural points. As shown in Fig. 7 in the main body, the purple ball in the top right corner is moved to a distance. This is the simplest way to edit the scene. Because all information is stored in the point representation.

**Object recoloring**. Our representation also supports recoloring. One example is exhibited in Fig. 7 in the main paper. In this instance, We replace the red ball's color and low-level appearance features ($F_c$ and $F_l$ in Eq. 5) with those in the left purple one while maintaining its high-level geometry features. In addition, if the number of points of the two objects is not identical, we compute the feature in the second object by interpolating the neighbor points in the original object. We simply use the inverse distance weights to obtain the interpolated feature vectors, this process is shown in Eq. 14.

$$f_a(x) = \frac{\sum_{k=1}^{N} w_k f_k}{\sum_{k=1}^{N} w_k}$$
$$w_k = \frac{1}{||x - p_k||}$$

(14)

where $N$ is the number of neighbors, in our experiments we select 8 as $N$. $x$ refers to the point in the second object. $p_k$ denotes the neighboring points in the first object.

**Object deformation**. We stretch the stool in the drums scene in NeRF Synthetic Dataset, which can be seen in Fig. 7 in the main paper. To avoid nonuniform deformation of the point cloud, we first reconstruct the mesh representation from the original point cloud. For each vertex in the mesh, we assign the geometry and appearance features to it by Eq. 14. Then we use Blender to stretch the mesh. After the deformation occurs in the mesh, we sample points uniformly on the surface of the mesh and interpolate features from the mesh vertex. Here we obtain the deformed neural point field that can be used to render views. For the Lego example, we simply construct the mesh by connecting the points with their neighbors. Next, we twist the joint of the Lego model by Blender to get the deformed positions of points. While the geometry and appearance features remain the same as before deformation occurs. We can see that there is no obvious decrease in the rendering qualities between the original images and the deformed images.

## G    COMPARISON WITH OTHER POINT-BASED RENDERING METHODS.

Some existing methods combine point cloud with NeRF to conduct point-based rendering, represented by PointNeRF and Point2Pix. Our approach substantially differs from other point-based rendering methods. Works represented by PointNeRF focus on improving rendering quality and speed with the assistance of point clouds. Those denoted by Point2Pix focus on rendering novel views from colorful point clouds. However, our goal is to train a model that can generate neural representations from multiview images generalizing to different scenes. In this section, we compare the three approaches in detail. Table 8 gives information about the quantitative results for comparing the three approaches.

| Training Setting | Methods | NeRF Synthetic | | | DTU | | |
|---|---|---|---|---|---|---|---|
| | | PSNR↑ | SSIM↑ | LPIPS↓ | PSNR↑ | SSIM↑ | LPIPS↓ |
| Generalization | PointNeRF | 6.12 | 0.18 | 0.88 | 23.18 | 0.87 | 0.21 |
| | Point2Pix | 19.23 | 0.787 | 0.542 | 16.74 | 0.655 | 0.558 |
| | **Ours** | **29.31** | **0.960** | **0.081** | **27.67** | **0.945** | **0.118** |
| Finetuning | PointNeRF | 30.71 | 0.961 | 0.081 | 28.43 | 0.929 | 0.183 |
| | Point2Pix | 25.62 | 0.915 | 0.133 | 24.81 | 0.894 | 0.209 |
| | **Ours** | **33.28** | **0.983** | **0.037** | **31.65** | **0.970** | **0.081** |

Table 8: Quantitative Comparisons with point-based methods on NeRF Synthetic and DTU datasets.

### G.1    COMPARISON WITH POINTNERF

Our method is a novel point-based paradigm for generalizable NeRF reconstruction, whereas PointNeRF focuses on single-scene optimization. PointNeRF also requires pertaining to the DTU dataset, while it can be seen as an initialization and cannot generalize to unseen scenes.

In Fig. 20 we give an example to illustrate this. From this figure and Table 8, PointNeRF performs well on the DTU dataset but collapses on the NeRF dataset due to its pre-training on the DTU dataset. This lack of generalization ability is a significant drawback for PointNeRF. In contrast, our proposed model, although also trained on the DTU dataset, exhibits a strong generalization capability on out-of-distribution data. This ability to generalize well to unseen data is a key advantage of our method.

Additionally, we note that Point2pix, another baseline model, tends to produce blurry images. This limitation hampers the visual quality and fidelity of the rendered images.

Point-based rendering methods can generally be divided into two main steps: "image-to-point feature fetching" and "point-to-ray feature aggregation." We will outline their distinctions in terms of the two steps.

In the "image to point" step:

1.PointNeRF fetches features for points solely from a single image, but our method considers multiple-view images simultaneously, incorporating a broader range of visual information. 2. Additionally, our method explicitly takes occlusions into account during feature fetching. 3. Furthermore, while PointNeRF employs a single latent vector as the neural feature for each point, our method utilizes a pair of latent vectors to separately represent shape and appearance. 4. Our method leverages low-level features to capture color information and high-level features for geometry regression, PointNeRF only utilizes high-level features.

In the "point to ray" step:

1. PointNeRF employs uniform sampling to sample points, whereas our method fully utilizes geometry priors in the point cloud by employing a log sampling strategy. This enables our method to render more efficiently and achieve superior geometries. 2. PointNeRF simply aggregates features from the neural point cloud to the sampling points along rays using inverse distance weights. In contrast, our method introduces a learnable kernel that enables feature aggregation based on visibilities, improving the effectiveness of the aggregation process.

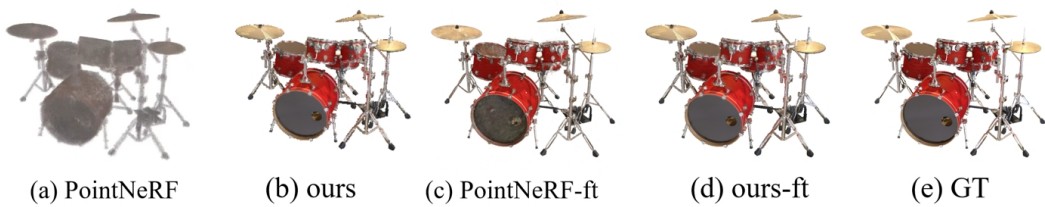

(a) PointNeRF        (b) ours        (c) PointNeRF-ft        (d) ours-ft        (e) GT

Figure 20: Comparisons with PointNeRF. ft refers to the per-scene optimization results.

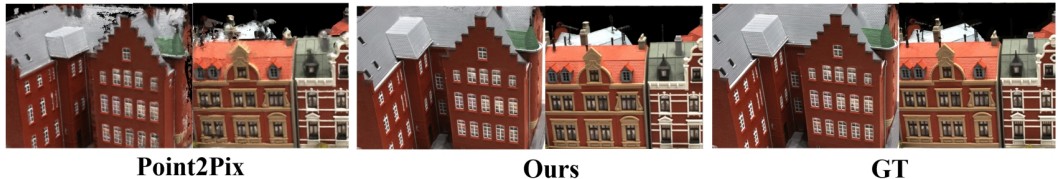

**Point2Pix**                    **Ours**                    **GT**

Figure 21: Comparison with Point2Pix.

## G.2 COMPARISON WITH POINT2PIX

Point2pix is proposed to render novel views from colorful point clouds rather than reconstructing neural radiance fields from multiview images, which is different from our methods. However, it can be categorized as point-based rendering. Therefore we also compare our method with it in terms of rendering qualities. We reimplement Point2pix with Pytorch and test it under our dataset configurations because there is no source code available at this moment. The qualitative results are shown in Fig 21. The point2pix cannot correctly render views at scenes with complex geometries and locally shape-varying areas. In addition, it is adversely affected by the imperfect point cloud, resulting in large holes and distortions in renderings. However, thanks to the point completion and finetuning in our hierarchical finetuning scheme, we can faithfully fill up holes and iteratively refine point clouds to achieve much better reconstructions. Moreover, our log sampling and learnable kernel modules enable us to perform well at geometric discontinuities.

