# OpenReview forum: "Learning Robust Generalizable Radiance Field with Visibility and Feature Augmented Point Representation"
_ICLR.cc/2024/Conference — ICLR 2024 poster_

### Official Review · Reviewer_gQRV · 2023-10-22

**Soundness:** 3 good
**Presentation:** 3 good
**Contribution:** 2 fair
**Rating:** 6
**Confidence:** 4

**Summary:**

This paper proposes to construct the generalizable neural field, called the generalizable neural Point Field (GPF), based on point-based rendering. This approach explicitly models by geometric priors and augments it with neural features to eliminate occlusions in feature-fetching. A nonuniform log sampling strategy is proposed to improve both rendering speed and reconstruction quality. Moreover, this paper presents a learnable kernel spatially augmented with features for feature aggregations, mitigating distortions at places with drastically varying geometries. Experiments show that the proposed model can deliver better geometries, view consistencies, and rendering quality on three datasets in both generalization and finetuning settings.

**Strengths:**

This paper proposes a Generalizable neural Point Field (GPF) for building generalizable NeRF based on point-based neural rendering. This paradigm outperforms existing image-based benchmarks and yields state-of-the-art performance on generic reconstructions.

This method explicitly models the visibilities by geometric priors and augments it with neural features, which are then used to guide the feature fetching procedure to better handle occlusions.

A nonuniform log sampling strategy is proposed based on the point density prior, and perturbations to sampling parameters are imposed for robustness, which not only improves the reconstructed geometry but also accelerates the rendering speed.

A spatially feature-augmented learnable kernel as feature aggregators is presented, which is proven to be effective for generic abilities and geometry reconstruction at drastically shape-varying areas.

**Weaknesses:**

This reviewer has the following concerns.

The primary contribution claimed by this paper is the introduction of the first generalizable NeRF based on point-based neural rendering. An existing point-based method is PointNeRF, which is a per-scene optimization method.
Section F.1 discusses the comparison between GPF and PointNeRF, which is helpful. What does the entry "Ours" represent in the table below Figure 17? If it refers to the model after fine-tuning, what are the results without fine-tuning?

Upon closer examination of the comparison with PointNeRF, this paper states that the improvement is attributed to hierarchical fine-tuning strategies. I am wondering about the runtime required for fine-tuning. PointNeRF can be optimized from scratch in approximately 40 minutes. Additionally, is pretraining of the generalizable NeRF necessary? It seems that the primary advantage of this method over PointNeRF lies in its superior fine-tuning strategy.

The log sampling strategy is simply a handcrafted sampling distribution around the surface, which may not be considered a significant technical contribution. This strategy can only be applied if the depth prior is known.

It is important to discuss the comparison between the proposed method and the recently introduced efficient Gaussian-splatting representation. What are the advantages of the proposed method?

**Questions:**

Please clarify the comparison with PointNeRF and explain the key factors that make this method superior to PointNeRF.

Please discuss the comparison between this method and Gaussian-splatting.

---

> ### Author Response · Authors · 2023-11-20
>
> >**Weakness 1**: The primary contribution claimed by this paper is the introduction of the first generalizable NeRF based on point-based neural rendering. An existing point-based method is PointNeRF, which is a per-scene optimization method. Section F.1 discusses the comparison between GPF and PointNeRF, which is helpful. What does the entry "Ours" represent in the table below Figure 17? If it refers to the model after fine-tuning, what are the results without fine-tuning?
>
> Thanks for the reviewer pointing out this. We apologize for the lack of the caption of the Table below **Figure 17** in the original paper. All metrics in the Table are tested under the finetuning setting. Following the reviewer’s suggestion, we additionally report the comparison of the results under the generalization settings of the three point-based rendering methods. We consolidate all the experimental results related to point-based methods into a single Table, i.e. **Table 9** in the revised Supplementary Material. For the convenience of the reviewer, we also provide the Table as follows:
>
> |Training  Setting&nbsp;| Methods   | NeRF Synthetic | DTU|
> |----------|---------------|-----------|----------|
> |           |    |PSNR↑         &nbsp; SSIM↑  &nbsp;  LPIPS↓   | PSNR↑    &nbsp;  SSIM↑   &nbsp; LPIPS↓       |
> || PointNeRF &nbsp;| 6.12    &nbsp;  &nbsp; &nbsp; 0.18   &nbsp;  &nbsp; &nbsp; 0.88         | 23.18       &nbsp; &nbsp; 0.87       &nbsp; &nbsp; &nbsp; 0.21  |
> |Generalization | Point2Pix    |19.23 &nbsp; &nbsp; 0.787   &nbsp; &nbsp; 0.542  | 16.74     &nbsp; &nbsp; 0.655    &nbsp; &nbsp; 0.558 |
> ||  **Ours**     |   **29.31**       &nbsp; &nbsp;   **0.960**   &nbsp; &nbsp;   **0.081**    |   **27.67**    &nbsp; &nbsp;   **0.945**       &nbsp; &nbsp;   **0.118**   |
> |                 | PointNeRF | 30.71    &nbsp; &nbsp; 0.961    &nbsp; &nbsp; 0.081      | 28.43  &nbsp; &nbsp; 0.929  &nbsp; &nbsp; 0.183  |
> |Finetuning| Point2Pix | 25.62      &nbsp; &nbsp; 0.915     &nbsp; &nbsp; 0.133       | 24.81  &nbsp; &nbsp; 0.894  &nbsp; &nbsp; 0.209  |
> || **Ours**      |  **33.28**           &nbsp; &nbsp;  **0.983**           &nbsp; &nbsp;  **0.037**          |  **31.65**    &nbsp; &nbsp;  **0.970**    &nbsp; &nbsp;  **0.081** |
>
> The results illustrate that the PointNeRF performs well on the DTU dataset but it is completely collapsed on the NeRF dataset because it is pre-trained on the DTU. This means that PointNeRF does not have any generalization ability to out-of-distribution data, as stated in **Figure 20** in the Appendix. In contrast, our model also trained on the DTU dataset, but can generalize very well on the out-of-distribution data. Moreover, Point2pix only produces blurry images.

---

> ### Author Response · Authors · 2023-11-20
>
> >**Weakness 2**: Upon closer examination of the comparison with PointNeRF, this paper states that the improvement is attributed to hierarchical fine-tuning strategies. I am wondering about the runtime required for fine-tuning. PointNeRF can be optimized from scratch in approximately 40 minutes. Additionally, is pretraining of the generalizable NeRF necessary? It seems that the primary advantage of this method over PointNeRF lies in its superior fine-tuning strategy.
> >**Question 1**: Please clarify the comparison with PointNeRF and explain the key factors that make this method superior to PointNeRF.
>
> Thanks for this comment. We here explain the differences between the PointNeRF and our method in detail and answer the reviewer's issue.
>
> In terms of optimization time, we would like to note that the duration of 40 minutes for PointNeRF does not refer to training the model from scratch. Rather, it corresponds to the fine-tuning process using a pretraining checkpoint obtained from the DTU dataset. This information can be verified by referring to the official code repository of PointNeRF (https://github.com/Xharlie/pointnerf).
>
> Below are the results of our evaluation, where we compared the fine-tuning time of our method with that of PointNeRF. The test set used for this comparison was scan114 of the DTU dataset, specifically in the context of fine-tuning.
>
> | Step (PSNR) | 1k | 10k | 15k |
> |------------|:---------------:|:-----------:|:----------:|
> |PointNeRF  | 2min (28.43)   | 20min (30.01) | 30min (30.1) |
> |**Ours**       | 1.5min (29.67) | 15min (31.18) | 22.5min (31.7) |
>
> The Table reveals that our approach exhibits faster convergence and attains higher upper bounds compared to PointNeRF. This outcome can be attributed to our log-sampling strategy, which introduces robust geometric priors during the training process and reduces the number of sampling points. As a result, our method benefits from stronger geometric constraints, leading to improved performance in terms of convergence speed and achieving higher-quality results.
>
>
> The finetuning strategy is indeed a big difference. However, our approach encompasses other significant components that are dominant to our superior performance, especially the generalization ability. Point-based rendering methods can generally be divided into two main steps: "image-to-point feature fetching" and "point-to-ray feature aggregation." We will outline their distinctions in terms of the two steps.
>
> **There are *6* distinct differences with PointNeRF, also with most other point-based methods.**
> For the “image to point” step :
> ***1.***	PointNeRF fetches features for points solely from a single image, but our method considers
> multiple-view images simultaneously, incorporating a broader range of visual information.
> ***2.***	Moreover, our method explicitly takes occlusions into account during feature fetching, which PointNeRF does not consider.
> ***3.***	In addition, hile PointNeRF employs a single latent vector as the neural feature for each point, our method utilizes a pair of latent vectors to separately represent shape and appearance.
> ***4.***	Our method leverages low-level features to capture color information and high-level features for geometry regression, whereas PointNeRF only utilizes high-level features.
>
> For the “point to ray” step:
> ***1.***	employs uniform sampling to sample points, whereas our method fully utilizes geometry priors in the point cloud by employing a log sampling strategy, allowing our method to render faster and obtain better geometries.
> ***2.***	PointNeRF simply aggregates features from the neural point cloud to the sampling points along rays using inverse distance weights. In contrast, our method introduces a learnable kernel that enables feature aggregation based on visibilities, improving the effectiveness of the aggregation process.
>
> Each of the aforementioned components plays a critical role in the success of the Generalizable Point Field approach. Taking into account the response provided to the reviewer, we have restructured **Section G1** in the revised supplementary material to enhance clarity and readability. The revised section aims to provide a clearer and more comprehensive explanation of the divergences between the two approaches, facilitating a better understanding of the proposed method.

---

> ### Author Response · Authors · 2023-11-20
>
> >**Weakness 3**: The log sampling strategy is simply a handcrafted sampling distribution around the surface, which may not be considered a significant technical contribution. This strategy can only be applied if the depth prior is known.
>
> As the reviewer state that the log sampling strategy indeed relies on the depth prior. Therefore we propose a way to extract depth priors from the point cloud (introduced in **Section 3.1 Equation 1~4**) initialized by the MVS techniques and taking into account visibilities. To make optimal use of the extracted depth priors, we propose the implementation of log sampling. This technique offers dual benefits: it enhances rendering speed and elevates the quality of reconstructed geometries.
>
> The efficacy of log sampling is substantiated through the presentation of results in **Figure 14**, **Figure 16**, and **Table 3**. **Figure 14** and **Table 3**describes the efficiency and rendering quality among different sampling strategies, **Figure 16** are enlarged details of renderings that affected by the sampling methods. The visual evidence in **Figure 14** demonstrates the acceleration achieved through log sampling, while **Figure 16** showcases the improved fidelity of the reconstructions. Furthermore, the quantitative analysis provided in **Table 3** solidifies the functional superiorities of log sampling.
>
> Currently, our proposed method still relies on the initialization module, which employs MVS techniques to predict a point cloud from input images. However, to reduce this dependency and enhance the flexibility of our approach, we plan to introduce an alternative neural module to replace the MVS-based initialization. The alternative module could be jointly trained with our GPF to produce point initialization for the given scenes. Moreover, we present this prospect idea in detail in Section H of the revised manuscript and we hope this would address concerns of the reviewer regarding deep priors.

---

> ### Author Response · Authors · 2023-11-20
>
> >**Weakness 4**:  It is important to discuss the comparison between the proposed method and the recently introduced efficient Gaussian-splatting representation. What are the advantages of the proposed method?
> >**Question 2**:  Please discuss the comparison between this method and Gaussian-splatting.
>
> We thank the reviewer for the comment. 3D Gaussian Splatting (3DGS) recently has gained lots of attention due to its promising progress. 3DGS is a rasterization technique described in [1] that allows real-time rendering of photorealistic scenes learned from small samples of images. To summarize, the 3DGS focuses on **per-scene** fast optimization and real-time rendering, whereas our approach pays attention to the **generalization** to unseen scenes without retraining. The abilities of per-scene optimization between the two methods are similar, for example, on the NeRF dataset, both our method and 3DGS achieve 33.3 PSNR. We properly mention the concurrent work in our Introduction and Related work.
>
> [1] Kerbl, Bernhard, et al. "3D Gaussian Splatting for real-time radiance field rendering." ACM Transactions on Graphics (ToG) 42.4 (2023): 1-14.

---

> > ### Comment · Reviewer_gQRV · 2023-11-22
> >
> > I would like to express my gratitude to the authors for their comprehensive response. The new results and discussions effectively address most of my previous concerns. As a result, I will raise my rating.

---

> > > ### Author Response · Authors · 2023-11-22
> > >
> > > We are pleased to hear that the reviewer gQRV has raised the rate! We sincerely appreciate you for your kind support and all the valuable comments again.

---

### Official Review · Reviewer_s3fi · 2023-10-23

**Soundness:** 2 fair
**Presentation:** 2 fair
**Contribution:** 2 fair
**Rating:** 6
**Confidence:** 3

**Summary:**

The paper presents a novel paradigm in generalizable Neural Radiance Field (NeRF) research by introducing the Generalizable Neural Point Field (GPF). Unlike traditional NeRF methods that utilize image-based rendering, this work focuses on point-based rendering. The paper claims to address several prevalent issues with the existing image-based methods, including occlusion-related problems, artifacts, and performance drop-offs with varying view distances.

**Strengths:**

- Originality: The Generalizable Neural Point Field (GPF) is a fresh perspective in NeRF research, emphasizing point-based over image-based rendering.
- Quality: The proposed methods exhibit high-quality research and innovation, from the nonuniform log sampling strategy to the feature-augmented learnable kernel.
- Clarity: The paper's overall structure is clear, presenting a logical flow of ideas and discussions.
- Significance: If the claims are validated further, this research could serve as a benchmark in the NeRF domain.

**Weaknesses:**

- Lack of Detailed Explanations: Some sections, especially the technical components, could use more in-depth explanations or visual aids.
- Dependency on Other Technologies: The initial dependency on PatchmatchMVS might limit the paper's approach from being a standalone solution.
- Limited Experimentation: Testing on only three datasets might not showcase the full potential or limitations of the method.
- Complexity: The approach's intricate nature might pose scalability or efficiency challenges that haven't been addressed comprehensively.

**Questions:**

- Could the authors expand on the visibility-oriented feature fetching, possibly with diagrams, for better clarity? Figure (b) alone is not clear enough.
- Given the reliance on PatchmatchMVS for the initial point scaffold, how does this affect the scalability or deployment of GPF in diverse scenarios? The authors also mention in the limitation section that they want to propose a NeRF-based initialization module that can be trained from scratch. Please comment on how you plan to achieve this.
- Can the authors comment on potential efficiency challenges due to the method's complexity? For example, compare the rendering speed with E-NeRF.

---

> ### Author Response · Authors · 2023-11-20
>
> >**Weakness 1**: Lack of Detailed Explanations: Some sections, especially the technical components, could use more in-depth explanations or visual aids.
> >**Question 1**: Could the authors expand on the visibility-oriented feature fetching, possibly with diagrams, for better clarity? Figure (b) alone is not clear enough.
>
> We greatly appreciate the valuable suggestions provided. Considering the reviewer's advice, we have incorporated comprehensive and detailed illustrations and clarifications, accompanied by visual aids, to enhance the readers' comprehension and facilitate their understanding. In summary, the additional components include an in-depth explanation of the physical meaning of the visibility scores (**Section A1 and Figure 9**), detailed illustrations of different sampling strategies (**Section E and Figure 14**), and the low- and high-level feature aggregations (**Section A2**), an explanation of the initialization module (**Section E4 and H**), etc. Other important revisions are highlighted in red in the revised manuscript.

---

> ### Author Response · Authors · 2023-11-20
>
> >**Weakness 2**: Dependency on Other Technologies: The initial dependency on PatchmatchMVS might limit the paper's approach from being a standalone solution.
> >**Question 2**: Given the reliance on PatchmatchMVS for the initial point scaffold, how does this affect the scalability or deployment of GPF in diverse scenarios? The authors also mention in the limitation section that they want to propose a NeRF-based initialization module that can be trained from scratch. Please comment on how you plan to achieve this.
>
> We thank the reviewer for this question. Our method indeed needs the module to initialize a point scaffold, here we use the PatchmatchMVS but are not limited to it. Other multi-view methods can also be used such as Colmap or OpenMVG because our approach is not sensitive to the initial point cloud. If the initial point cloud is imperfect, our point growing, pruning, and refinement modules can make it better by optimization. We also plan to propose an alternative module to replace the currently used initialization module, which can be jointly trained with our GPF method. Here we briefly introduce what we are going to do for that.
>
> This module will be designed for initializing point clouds from multi-view images for reconstructing our generalizable point field. Therefore, we plan to use a feature-based method combined with a randomization process. We use a figure to visually aid in explaining this method, which is added to the Revised Appendix on **Page 23, Figure 22**. The process begins by uniformly sampling points from the given space. Next, we employ a hidden point removal algorithm [1] to roughly identify the points that are visible from the source viewpoints. During this step, only the spatial relationship between the viewpoints and the point set is considered, disregarding the scene content depicted in the images. Subsequently, we extract features from these images using a UNet-like network, as illustrated in **Figure 1(a)** and project these features onto their corresponding visible points.
>
>  As a result, each point contains multiple feature vectors. Following this, we either train another network or employ suitable metrics (such as cross-correlation) to evaluate the similarity of these features. If the features exhibit sufficient similarity, we can conclude that the given point lies on the surface of the object. Importantly, the features used for feature matching can also be reused for the reconstruction of the GPF, such as the low- or high-level features. Consequently, this approach does not impose an additional memory burden or introduce extra parameters.
>
> [1]  https://github.com/pdhimal1/HPR

---

> ### Author Response · Authors · 2023-11-20
>
> >**Weakness 3**: Limited Experimentation: Testing on only three datasets might not showcase the full potential or limitations of the method.
>
> We thank the reviewer for pointing out this issue. Following the suggestion of the reviewer, we conduct additional experiments for all baselines and our method on the LLFF dataset [1] for further comparisons. The qualitative and quantitative results are reported in Section C1 of the revised Appendix, including the new **Figure 14** and **Table 5**. To facilitate the reviewing process, we list the quantitative Table on the new dataset in the following:
>
> | Training Setting &nbsp;&nbsp; | Generalization                                 | Finetuning |
> |------------------|---------------------------------------------|--------------|
> | Methods          |  PSNR↑ &nbsp; SSIM↑ &nbsp; LPIPS↓ &nbsp; &nbsp;|  PSNR↑ &nbsp; SSIM↑ &nbsp; LPIPS↓ |
> | IBRNet           | 25.13  &nbsp; &nbsp;  0.817 &nbsp; &nbsp; 0.205 &nbsp; &nbsp;| 26.73 &nbsp; &nbsp;  0.851 &nbsp; &nbsp;  0.175  |
> | MVSNeRF          | 21.93  &nbsp; &nbsp; 0.795 &nbsp; &nbsp; 0.252  &nbsp; &nbsp;| 25.45 &nbsp; &nbsp;  0.877 &nbsp; &nbsp;  0.192  |
> | ENeRF            | 22.78 &nbsp; &nbsp;  0.808 &nbsp; &nbsp; 0.209 &nbsp; &nbsp;| 24.89 &nbsp; &nbsp;  0.865 &nbsp; &nbsp;  0.199  |
> | Neuray           | 25.35  &nbsp; &nbsp; 0.818 &nbsp; &nbsp; 0.198  &nbsp; &nbsp;| 27.06 &nbsp; &nbsp;  0.850 &nbsp; &nbsp;  0.172  |
> | **Ours**             | **26.01** &nbsp; &nbsp;  **0.829** &nbsp; &nbsp; **0.184**  &nbsp; &nbsp;|**27.79** &nbsp; &nbsp;  **0.872** &nbsp; &nbsp;  **0.171**  |
>
> The obtained results clearly indicate that our approach maintains its superiority in both the generalization and fine-tuning settings. we have consistently outperformed the alternative methods across various evaluation metrics.
>
> [1] Mildenhall, Ben, et al. "Local light field fusion: Practical view synthesis with prescriptive sampling guidelines." ACM Transactions on Graphics (TOG) 38.4 (2019): 1-14.

---

> ### Author Response · Authors · 2023-11-20
>
> >**Weakness 4**: Complexity: The approach's intricate nature might pose scalability or efficiency challenges that haven't been addressed comprehensively.
> >**Question 3**: Can the authors comment on potential efficiency challenges due to the method's complexity? For example, compare the rendering speed with E-NeRF.
>
> Thanks for this comment. We follow the reviewer’s advice to report some analysis and metrics reflecting the efficiency. First, we would argue that even if our method needs to calculate the visibility maps before rendering, these maps can be precomputed and reused many times. As a result, the rendering speed remains unaffected. Additionally, the computation of visibility maps is well-suited for parallel processing, allowing for rapid execution.
>
> Furthermore, we measure the time for all baselines and our method of rendering a single image on the DTU dataset by using a single NVIDIA 3090 GPU. We also calculate the number of parameters of these models (Param in the Table below). These provide insights into the complexity and resource requirements of the proposed method.
> We also provide the results and analysis on **Page 20** in the Highlighted Revision Version. Here for the convenience of the reviewer, we present the Table in the following as well:
>
> Method &nbsp; | IBRNet   &nbsp;     | MVSNeRF &nbsp;  | ENeRF  &nbsp;   | Neuray &nbsp;    | Ours  &nbsp;    |
> |----------|---------------|-----------|----------|-----------|----------|
> |Param  &nbsp; | 8.95e6       | 4.68e5   | 4.36e5   | 2.85e7   | 7.92e4   |
> |Speed (s)&nbsp;   | 30.85        | 4.963    | 0.34     | 35.18    | 0.83   |
>
>
> It can be seen that our method exhibits the second-fastest rendering speed, with only a slight difference compared to ENeRF. However, despite ENeRF having a larger parameter count, it achieves a slightly faster rendering speed. This is primarily due to ENeRF compressing the number of sampled points to an extreme reduction of 2, whereas our approach employs log sampling with 16 sampling points. The impact of this influencing factor can be observed in **Figure 14** in the Appendix.
>
> Another reason is that the k-nearest search algorithm also consumes a few times. In addition, one advantage of our method is the ability to freely adjust between rendering speed and quality. Because the rendering speed is sensitive to the number of points due to the K nearest search operation, more points typically result in better rendering quality but slower rendering speed, and vice versa. In the above experiments, we maintain 200,000 points.

---

> > ### Comment · Reviewer_s3fi · 2023-11-22
> > **Thank you!**
> >
> > Thank you so much for your efforts in addressing my concerns. I have no further questions and would like to stick to my initial positive opinion.

---

> > > ### Author Response · Authors · 2023-11-22
> > >
> > > Once again, thank you very much for your insightful comments and suggestions on our paper, which largely makes it more rigorous and comprehensive.

---

### Official Review · Reviewer_BQBs · 2023-11-02

**Soundness:** 2 fair
**Presentation:** 2 fair
**Contribution:** 3 good
**Rating:** 6
**Confidence:** 4

**Summary:**

The work propose a generalizable point-based NeRF for novel-view synthesis tasks. The point cloud is first initalized from classical MVS technique. A U-Net is trained to extract feature for the input views, which is aggregated into the points in a visibility-aware and learnable manner. To do volume rendering, query coordinates are sampled per the point density to improve efficiency. The query coordinates search K-nearest neighbor from the feature point cloud to form the query point feature, which is then map to density and color for volume rendering.

**Strengths:**

The quantitative improvement is solid. I believe many of the proposed modules can be plug into point-based rendering system to boost results.

**Weaknesses:**

The visibility score in Eq4 is not well designed. The score actually decay more quickly for point ahead of the depth ($P_z < D_{i,xy}$). Consider two points with $P_z^{(back)} = D_{i,xy} + \epsilon$ and $P_z^{(front)} = D_{i,xy} - \epsilon$, their scores are:
- $score^{(back)} = 1 - \frac{|D_{i,xy} + \epsilon - D_{i,xy}|}{D_{i,xy} + \epsilon} = \frac{D_{i,xy}}{D_{i,xy} + \epsilon}$
- $score^{(front)} = 1 - \frac{|D_{i,xy} - \epsilon - D_{i,xy}|}{D_{i,xy} - \epsilon} = \frac{D_{i,xy} - 2\epsilon}{D_{i,xy} - \epsilon}$

When $\epsilon > 0$, $score^{(front)} < score^{(back)}$. In addition, the score is claimed to be naturally constrainted in range of 0 and 1, but it is not the case when $P_z < 0.5 D_{i,xy}$ (become negative).

**Questions:**

I found some qualitative results from the baseline is better. In Fig3, the head of the ship and the sea in the ship scene, the table of the durian scene are better recovered by ENeRF. In Fig4, the reconstructed ground dirt in the BlendedMVS is overly smooth while the baseline ENeRF can recover more detail texture. What would be the reason of these? Is it because the dependent MVS point cloud?

Is the proposed method sensitive to the initial points?

Paper proofread:
- Missing parentheses for the $\exp$ in Sec3.3 last paragraph.
- The reference to the Figure in Sec.4.2's 5th sentence is missing.

---

> ### Author Response · Authors · 2023-11-20
>
> >**Weakness**: The visibility score in Eq4 is not well designed.
>
> Thanks for the comment. We sincerely apologize for misleading you due to our lack of detailed description. Actually, the $P_z$ in **Eq. 4** is naturally larger than $D_{i,xy}$ due to the definition of depth projection. In order to enhance clarity and facilitate comprehension of this matter, we have incorporated an additional **Figure 9** in the revised Supplementary Material to serve as an illustrative tool.  The variable $P_z$ denotes the z-value associated with a specific point in the camera coordinate system. It represents the length of the projection of the line connecting the point and the camera center onto the camera's optical axis ($P_{z,c}$ in the figure). The value of $D_{i,xy}$ is derived by projecting the point onto the depth map and subsequently employing bilinear interpolation. The depth map itself is computed by **Equation 1 to 3** in the main paper. These relationships are visually depicted in the aforementioned Figure.
>
> If a point, such as point A in the figure, is
> visible from the viewpoint, it implies that it lies on the object’s surface. In this case, the z-value of
> the point in the camera coordinate system should equal the projected depth value, $D_{i,xy}$.
> On the contrary, if a point (e.g., point B) is not visible from the viewpoint, its z-coordinate can only
> be greater than $D_{i,xy}$. There should not be any other points between $D_{i,xy}$, and the camera’s center,
> as it would cause a change in the value of $D_{i,xy}$ accordingly. Consider another case where a point
> lies outside the viewing angle’s frustum, and its z-value is smaller than the minimum depth on the
> depth map, as illustrated by point C in the figure. In this case, the point would be projected outside
> the image plane. However, during interpolation on the plane, we employ zero padding. Therefore,
> the value of $D_{i,xy}$ for this point would be zero, which is smaller than $P_z$ as well.
>
> **To conclude, the $P_z$ is always greater than or equal to $D_{i,xy}$**. Therefore, the decay rate would be stable, and the score would be constraint to [0, 1]. We hope the above interpretaion could dispel the reviewer's concen.
>
> In order to enhance the readability and comprehensibility of the manuscript, we have incorporated the aforementioned additional descriptions into **Section A1** of the revised Appendix.

---

> ### Author Response · Authors · 2023-11-20
>
> >**Question**: I found some qualitative results from the baseline is better. In Fig3, the head of the ship and the sea in the ship scene, the table of the durian scene are better recovered by ENeRF. In Fig4, the reconstructed ground dirt in the BlendedMVS is overly smooth while the baseline ENeRF can recover more detail texture. What would be the reason of these? Is it because the dependent MVS point cloud?
> Is the proposed method sensitive to the initial points?
>
> We appreciate the reviewer for bringing this to our attention. Indeed some examples generated by ENeRF exhibit clearer background objects, such as the sea in the "ship" example and the table in the "durian" example. We would like to provide two arguments in response to this observation.
>
> On the one hand, the point scaffold of our approach is initialized by the depth fusion algorithm from the predicted depth maps. As a result, the density of the point cloud belongting to the primary objects (i.e., the ship and the durian themselves) tends to be higher compared to that of the background objects (i.e., the sea and the table). This is because the camera views primarily focus on capturing the main subject, while there might only be a few camera views that capture the background objects. Consequently, the rendering quality of the background objects might be slightly inferior to that of the main object in a given scene. Improving the density of the point clouds in these areas can enhance the corresponding qualities.
>
> On the other hand, ENeRF and other image-based methods select source views that are spatially close to the target views for rendering. These source images contain a comparable amount of information regarding both the background and the main subject. Hence, sometimes they could obtain clearer background objects. This is also a reason why our approach can effectively recover the main objects in a scene for both the rendering quality and the geometry, while the baselines cannot faithfully reconstruct the main objects.
>
> In addition, as for the over-smoothness of the “excavator” scene in BlenededMVS, this can be attributed to the insufficient number of initial points used. In this case, we only utilize 200k points to initialize the scene, which is inadequate considering the vastness of the scene. Consequently, the point density in space becomes low, making it challenging to effectively represent high-frequency information in the scene and resulting in image smoothness.
>
> This can be alleviated by more initial points or a smaller search radius in the finetuning setting to grow up more points.  While our images may appear smooth with a small number of points, our method successfully reconstructs more comprehensive geometric structures compared to the baseline methods, particularly in heavily occluded areas.
>
> Besides, we incorporate this valuable comment into our Appendix. We provide an additional example which is produced from 400k initial points and smaller search radius in the new **Figure 19** of the revised Appendix. This example successfully recovers most of the high-frequency information in the scene by using more initial points. We think this should be a good example to illustrate the effect of the number of points.
>
> In conclusion, we figure that our approach is **not** sensitive to the initialization of point clouds due to our point growing, point pruning, and point refinement modules, which can be seen in **Section 4.3** of the main paper. Although the initial point clouds are not so perfect, we can still optimize them with these techniques.

---

### Official Review · Reviewer_BFzR · 2023-11-04

**Soundness:** 3 good
**Presentation:** 3 good
**Contribution:** 3 good
**Rating:** 8
**Confidence:** 4

**Summary:**

This paper proposes a novel paradigm for constructing a generalizable neural field based on point-based rendering, which addresses the challenges of occlusions, distortions, and degradation in image-based representations. The proposed approach combines geometric priors and neural features to eliminate occlusions in feature-fetching, and a nonuniform log sampling strategy and a learnable kernel spatially augmented with features for improved rendering speed and reconstruction quality. The authors demonstrate the effectiveness of their approach on a variety of datasets, showing improved generalization and robustness to occlusions and distortions compared to previous methods. Overall, the paper presents a promising approach for learning robust and generalizable radiance fields.

**Strengths:**

Overall the paper is nicely presented and introduce several novel components including:

- The proposed approach combines geometric priors and neural features to eliminate occlusions in feature-fetching explicitly in the rendering process, which is a novel contribution to the field.
- The authors introduce a nonuniform log sampling strategy and a learnable kernel spatially augmented with features, which is a novel approach to improving rendering speed and reconstruction quality.

The method is properly evaluated with some ablation study.

**Weaknesses:**

The paper proposed a new pipeline with several novel designs over the existing methods, however many of the designs are not validated and some of the claims are not strongly backed by their existing experiments:
1. The exact definition of convergence speed is vague in Table 2, and is not explained in details. Making the results in this table questionable.
2. Separating low level and high level features sounds intuitive but however is not validated and the necessity of such design is thus questionable.
3. No quantitative validation on claims such as "better geometry" and "occlusion awareness".

Also the paper needs more work to reduce typos and fix the format.

**Questions:**

1. Visualize the test PSNR curve instead of just stating "Convergence Speed" as in Table 2 would be more convincing, intuitive and easier to follow.
2. Some modules are not well ablated and analyzed - \eg high-low-level feature encoding.
3. It would be nice to present some metric and quantitatively prove the quality in geometry (esp. occlusion awareness). At least depth error should be compared with MVSNeRF and NeuRay.
4. In related works, Generalizable Neural Field. section: "All the above can be seen as image-based neural rendering". I think this might be inaccurate- I believe the finetuned/unfinetuned MVSNeRF / GeoNeRF / NeRFusion can aggregate multi view information and do not require original images for further use (though MVSNeRF fetches image color for rendering in some versions). Could you clarify on this? Also I believe the section is not extensive enough. The authors should also talk specifically about other point-based neural rendering methods, maybe in a dedicated section.
5. Maybe: considering other point-based methods as ft baselines and include in the main paper.
6. Typos and minor fixes:
  - Table 2: Convergeuce -> Convergence; Missing/misplaced underline under 1.04s
  - Citation format should be fixed throughout the paper.

---

> ### Author Response · Authors · 2023-11-20
>
> > **Weakness 1**: The exact definition of convergence speed is vague in Table 2, and is not explained in details. Making the results in this table questionable.
> >**Question 1**: Visualize the test PSNR curve instead of just stating "Convergence Speed" as in Table 2 would be more convincing, intuitive and easier to follow.
>
> We appreciate the reviewer for bringing this to our attention. We have taken the feedback into account and have made the necessary revisions. In the revised Appendix, we have included a new figure, **Figure 14**, to provide a comprehensive comparison of efficiency among all counterparts. The horizontal axis represents training time, the vertical axis represents PSNR, and the size of the circles represents the time required for rendering an 800*800 image.
>
> Upon examining this figure, it becomes evident that the log sampling method requires the least training time while achieving the highest PSNR. Furthermore, it exhibits the second-fastest rendering speed, with only a slight slow compared to the "surface sampling". To avoid any potential misunderstanding, we have also updated the "convergence speed" and "rendering speed" in **Table 3** (original Table 2) of the manuscript to "training time" and "rendering time" respectively.

---

> ### Author Response · Authors · 2023-11-20
>
> >**Weakness 2**: Separating low level and high level features sounds intuitive but however is not validated and the necessity of such design is thus questionable.
> >**Question 2**: Some modules are not well ablated and analyzed - \eg high-low-level feature encoding.
>
> Thanks for this constructive comment. The motivation behind separating low- and high-level features in our approach draws inspiration from the theory of convolution. In CNN, the initial layers tend to capture low-level features related to color and corners, while deeper layers focus on extracting high-level, semantic, and abstract features. This observation has guided our decision to differentiate the use of low-level and high-level features in our method. Furthermore, research related to MVS, such as MVSNET [1] and MVSNeRF [2], has demonstrated that leveraging high-level features in feature-matching algorithms can lead to more accurate and reliable correspondences. Building upon this insight, we utilize low-level features for color regression and high-level features for decoding the $\sigma$ in NeRF.
>
>
> We agree with the reviewer that this design requires further clarifications and ablations. In response to this, we conducted experiments using the DTU dataset to analyze the effects of different feature combinations on rendering quality and reconstructed geometries. The experiment settings include "only low-level features used", "only high-level features used", and "the combinations of the low- and high-level features". To provide an example, when referring to "only low-level features," it means that both the color and density decoders receive low-level features as input to regress their respective outputs. This configuration allows us to specifically investigate the influence of utilizing solely low-level features on the rendered results. The ablation results are listed below:
>
> |Training Setting &nbsp; |Generalization    |Finetuning     | |
> |----------|---------------|-----------|----------|
> |Methods |PSNR↑   &nbsp;  SSIM↑   &nbsp; LPIPS↓   &nbsp;  &nbsp;  | PSNR↑    &nbsp;   SSIM↑   &nbsp;   LPIPS↓  &nbsp;  &nbsp;  | RMSE↓  &nbsp; &nbsp;  Acc.T↑    |
> |Only-low  | 24.15   &nbsp; &nbsp;  0.896   &nbsp; &nbsp;  0.178   | 30.16    &nbsp; &nbsp;  0.921    &nbsp; &nbsp;  0.133  |0.383  &nbsp; &nbsp;  0.865 |
> |Only-high  | 22.94   &nbsp; &nbsp;  0.852   &nbsp; &nbsp;  0.231  | 30.47    &nbsp; &nbsp;  0.928   &nbsp; &nbsp;  0.132   |0.155  &nbsp; &nbsp;  0.917   |
> |low-high  |   **27.67**    &nbsp; &nbsp;  **0.945**    &nbsp; &nbsp;   **0.118**   |   **31.65**     &nbsp; &nbsp;   **0.970**    &nbsp; &nbsp;     **0.081**     |     **0.122**   &nbsp; &nbsp;      **0.936** |
>
> It is noted that both of the counterparts suffer from performance degradation. In addition, even though the rendering quality of “only low-level features” is better than that of the “only high-level features”, it is worth noting that the reconstruction geometries in the former are comparatively worse. This insight reinforces the significance of incorporating both low-level and high-level features in our approach. We also add these additional ablations to **Section E2** of the revised Appendix.
>
> [1] Yao, Yao, et al. "Mvsnet: Depth inference for unstructured multi-view stereo." Proceedings of the European conference on computer vision (ECCV). 2018.
>
> [2] Chen, Anpei, et al. "Mvsnerf: Fast generalizable radiance field reconstruction from multi-view stereo." Proceedings of the IEEE/CVF International Conference on Computer Vision (ICCV). 2021.

---

> ### Author Response · Authors · 2023-11-20
>
> >**Weakness 3**: No quantitative validation on claims such as "better geometry" and "occlusion awareness".
> >**Question 3**: It would be nice to present some metric and quantitatively prove the quality in geometry (esp. occlusion awareness). At least depth error should be compared with MVSNeRF and NeuRay.
>
> We thank the reviewer for pointing out this issue. We highly agree with the reviewer that the quantitative metric for reconstruction quality is essential. In the revised **Section D**, we provide the RMSE an Accuracy with Threshold metrics to assess the quality of reconstructed depths for both the baseline methods and our own. Furthermore, as a means of validating the efficacy of explicit occlusion modeling, we present an alternate version of our approach, denoted as $Ours_{wo/vis}$, which excludes the visibility scores outlined in **Equation 5** of the main paper, marked as $Ours_{wo/vis}$. Here we provide this Table for the convenience of reviewing.
>
>
> |Dataset|	 Method          | IBRNet   | ENeRF  | Neuray  | Ours$_{wo/vis }$  | **Ours**  |
> |----------|---------------|-----------|----------|-----------|----------|----------|
> |NeRF Synthetic | RMSE↓ | 0.677   | 0.527    | 0.547    | 0.294       | **0.161**    |
> |  		      | Acc.T↑ | 0.380    | 0.159   | 0.119    | 0.596    | **0.787**   |
> |DTU          | RMSE↓  | 0.321  | 0.435   | 0.162    | 0.189   | **0.122**   |
> |        | Acc.T↑ | 0.896    | 0.741    | 0.911    | 0.905       | **0.936**     |
>
> The Acc. T denotes the accuracy of the threshold, we set the threshold as $1.25^3$ for all experiments. In addition, all tests are under generalization settings because finetuning cannot faithfully reflect the real understanding of models to the realistic geometries. Our approach, in particular, demonstrates superior results, particularly on the NeRF dataset. This indicates that our method is capable of capturing plausible geometries even when dealing with out-of-distribution data. Additionally, it should be highlighted that removing the visibility score can result in a decrease in performance, which validate the highly positive effect of explicitly modeling visibility on the generalization to unseen scenarios.

---

> ### Author Response · Authors · 2023-11-20
>
> >**Question 4**: In related works, Generalizable Neural Field. section: "All the above can be seen as image-based neural rendering". I think this might be inaccurate- I believe the finetuned/unfinetuned MVSNeRF / GeoNeRF / NeRFusion can aggregate multi view information and do not require original images for further use (though MVSNeRF fetches image color for rendering in some versions). Could you clarify on this? Also I believe the section is not extensive enough. The authors should also talk specifically about other point-based neural rendering methods, maybe in a dedicated section.
>
> We apologize for the lack of clarity in our previous statement regarding "image-based" rendering. To clarify, the **unfinetuned** versions of MVSNeRF and GeoNeRF still require all source images to synthesize novel views. While MVSNeRF employs the neural encoding volume to render images, it still need to select source images that are spatially close to the target view and reconstruct the volume before each time of rendering. On the other hand, as indicated by **Eq 3** in GeoNeRF, it still requires full-resolution 2D feature maps to render novel views, which prevents it from discarding any source images.
>
> By contrast, our proposed method is fully independent of the source images both in **both** generalization and finetuning settings. As stated by the reviewer, the **finetuned** MVSNeRF, GeoNeRF and NeRFusion are indeed independent of the source images when synthesizing novel views. However, it should be noted that these models rely on the neural volume, which is not directly accessible for manipulation, as we claimed in our Introduction. If one desires to manipulate and edit the neural volume, one would have to begin by manipulating the source images used for reconstructing the volume. To address this ambiguity, we have made the necessary modifications to the related statement on **Page 2** in the Highlighted Revision Version.
>
> Furthermore, we follow the reviewer's feedback to include the "Point-based Rendering" into our **Related Work** as a dedicated section on **Page 3** in the Highlighted Revision Version.

---

> ### Author Response · Authors · 2023-11-20
>
> >**Question 5**: Maybe: considering other point-based methods as ft baselines and include in the main paper.
>
> Thank you for your comment. We have included qualitative and quantitative comparisons with two distinct point-based rendering methods, namely PointNeRF [1] and Point2Pix [2], in the revised supplementary materials, **Pages 20 to 21**. These comparisons contain both generalization and finetuning settings to provide a comprehensive evaluation. We can see that both of the two methods cannot generalize well to unseen scenarios, especially PointNeRF. It only achieves 6 PSNR on the out-of-distribution data in the NeRF dataset. However, it is worth noting that PointNeRF exhibits favorable performance on the per-scene optimization task. Therefore, we have taken the reviewer's advice to include the finetuning results as the baselines to our main paper on **Page 7**.
>
> To make it easier for reviewing purposes, we have provided the table here.
>
>
> |Training  Setting&nbsp;| Methods   | NeRF Synthetic &nbsp;&nbsp;| DTU|
> |----------|---------------|-----------|----------|
> |           |    |PSNR↑         &nbsp; SSIM↑  &nbsp;  LPIPS↓   | PSNR↑    &nbsp;  SSIM↑   &nbsp; LPIPS↓       |
> || PointNeRF &nbsp;| 6.12    &nbsp;  &nbsp; &nbsp; 0.18   &nbsp;  &nbsp; &nbsp; 0.88         | 23.18       &nbsp; &nbsp; 0.87       &nbsp; &nbsp; &nbsp; 0.21  |
> |Generalization | Point2Pix    |19.23 &nbsp; &nbsp; 0.787   &nbsp; &nbsp; 0.542  | 16.74     &nbsp; &nbsp; 0.655    &nbsp; &nbsp; 0.558 |
> ||  **Ours**     |   **29.31**       &nbsp; &nbsp;   **0.960**   &nbsp; &nbsp;   **0.081**    |   **27.67**    &nbsp; &nbsp;   **0.945**       &nbsp; &nbsp;   **0.118**   |
> |                 | PointNeRF | 30.71    &nbsp; &nbsp; 0.961    &nbsp; &nbsp; 0.081      | 28.43  &nbsp; &nbsp; 0.929  &nbsp; &nbsp; 0.183  |
> |Finetuning| Point2Pix | 25.62      &nbsp; &nbsp; 0.915     &nbsp; &nbsp; 0.133       | 24.81  &nbsp; &nbsp; 0.894  &nbsp; &nbsp; 0.209  |
> || **Ours**      |  **33.28**           &nbsp; &nbsp;  **0.983**           &nbsp; &nbsp;  **0.037**          |  **31.65**    &nbsp; &nbsp;  **0.970**    &nbsp; &nbsp;  **0.081** |
>
> [1] Xu, Qiangeng, et al. "Point-nerf: Point-based neural radiance fields." Proceedings of the IEEE/CVF Conference on Computer Vision and Pattern Recognition (CVPR). 2022.
>
> [2] Hu, Tao, et al. "Point2Pix: Photo-Realistic Point Cloud Rendering via Neural Radiance Fields." Proceedings of the IEEE/CVF Conference on Computer Vision and Pattern Recognition (CVPR). 2023.

---

> > ### Comment · Reviewer_BFzR · 2023-11-23
> >
> > Thanks! I'm raising my score since all concerns have been clearly addressed.

---

> > > ### Author Response · Authors · 2023-11-23
> > >
> > > We appreciate your positive feedback very much! We incorporate most of your suggestions into our final version, which largely improve the paper’s quality, and we express gratitude for that again!

---

### Author Response · Authors · 2023-11-20
**Overall Responses**

We express our sincere appreciation to the reviewers for their valuable and insightful feedback. We sincerely thank the time and attention of the Area Chair throughout the review process. We especially appreciate reviewers for recognizing our paper as novel (BFzR
), improvement solid (BQBs), original and fresh (s3fi) and better performance (gQRV).

We apologize for the delay in our response, as we have been diligently working on conducting extensive supplementary experiments and making important revisions to the manuscript.

We have corrected all typos following reviewers' advice, enhanced the readability of the manuscript, and reorganized the structure of the Appendix. We provide responses to each reviewer's inquiries in the following section. We select and incorporate pertinent portions of these responses into the paper to enhance the overall quality and comprehensiveness of our manuscript. To facilitate the reviewer's understanding and identification of the significant modifications made to the paper, all crucial revisions have been highlighted in **red** within the revised manuscript.

---

### Author Response · Authors · 2023-11-22

Dear AC and Reviewers,

We appreciate again for all reviewers' valuable suggestions which have been very helpful in improving our paper. Please do not hesitate to contact us if you have further questions.

Since the disscussion is approaching the deadline (about 1 day), we wonder whether the reviewers' concerns have been resolved by our reply.

Best Regards,
Authors

---

### Meta-Review · Area_Chair_As35 · 2023-12-09

**Metareview:**

(a) The paper introduces a novel paradigm in Neural Radiance Field (NeRF) research, focusing on point-based rendering, termed Generalizable Neural Point Field (GPF). It addresses the challenges of occlusions, distortions, and degradation common in image-based representations. The approach innovatively combines geometric priors and neural features for occlusion-free feature-fetching and employs a nonuniform log sampling strategy alongside a learnable kernel, augmented with features to enhance rendering speed and reconstruction quality. The authors present empirical evidence demonstrating their method's superior generalization and robustness to occlusions and distortions across multiple datasets, outperforming previous methods.

(b) Strengths: The paper presents an innovative approach by focusing on point-based rendering, which is a significant departure from traditional image-based methods in NeRF research. It addresses key challenges in the field, such as occlusions and distortions, through a well-thought-out combination of geometric priors and neural features. The paper showcases solid quantitative improvements over existing methods, backed by comprehensive experiments.

(c) Weaknesses:
The original submission had vague definitions and claims, particularly regarding convergence speed and feature separation, that needed clarification.
Initially, there was a lack of quantitative validation for specific claims, such as "better geometry" and "occlusion awareness."

However, the authors have made good efforts to address the reviewers' concerns, providing additional experiments, detailed clarifications, and revisions to the manuscript.

**Justification For Why Not Higher Score:**

The original submission had vague definitions and claims, particularly regarding convergence speed and feature separation, that needed clarification. Initially, there was a lack of quantitative validation for specific claims, such as "better geometry" and "occlusion awareness." Even though the authors preliminarily addressed these concerns in the rebuttal, the issues indicate that the method might require further refinement and testing more rigorously.

**Justification For Why Not Lower Score:**

The paper presents an innovative approach by focusing on point-based rendering, which is a significant departure from traditional image-based methods in NeRF research. It addresses key challenges in the field, such as occlusions and distortions, through a well-thought-out combination of geometric priors and neural features. The paper showcases solid quantitative improvements over existing methods, backed by comprehensive experiments.

The authors have responded thoughtfully and thoroughly to all the concerns raised by the reviewers. They have provided additional data, detailed explanations, and significant revisions to their manuscript.

---

### Decision · Program_Chairs · 2024-01-16

Accept (poster)